# Engineering interfacial sulfur migration in transition-metal sulfide enables low overpotential for durable hydrogen evolution in seawater

Min Li[1,2], Hong Li[3], Hefei Fan[1], Qianfeng Liu [1], Zhao Yan [1], Aiqin Wang [3], Bing Yang [3] ✉ & Erdong Wang [1] ✉

Hydrogen production from seawater remains challenging due to the deactivation of the hydrogen evolution reaction (HER) electrode under high current density. To overcome the activity-stability trade-offs in transition-metal sulfides, we propose a strategy to engineer sulfur migration by constructing a nickel-cobalt sulfides heterostructure with nitrogen-doped carbon shell encapsulation (CN@NiCoS) electrocatalyst. State-of-the-art ex situ/in situ characterizations and density functional theory calculations reveal the restructuring of the CN@NiCoS interface, clearly identifying dynamic sulfur migration. The NiCoS heterostructure stimulates sulfur migration by creating sulfur vacancies at the $Ni_3S_2$-$Co_9S_8$ heterointerface, while the migrated sulfur atoms are subsequently captured by the CN shell via strong C-S bond, preventing sulfide dissolution into alkaline electrolyte. Remarkably, the dynamically formed sulfur-doped CN shell and sulfur vacancies pairing sites significantly enhances HER activity by altering the $d$-band center near Fermi level, resulting in a low overpotential of 4.6 and 8 mV at 10 mA cm$^{-2}$ in alkaline freshwater and seawater media, and long-term stability up to 1000 h. This work thus provides a guidance for the design of high-performance HER electrocatalyst by engineering interfacial atomic migration.

Hydrogen fuel with its high calorific value and clean combustion products, is recognized as an ideal energy carrier compared to other storage media[1,2]. Seawater has garnered considerable attention as a feedstock for the hydrogen evolution reaction (HER), crucial for large-scale hydrogen production due to limited freshwater resources[3,4]. However, challenges persist in seawater electrolysis, primarily stemming from HER electrode deactivation caused by chlorine corrosion, calcium ($Ca^{2+}$) and magnesium ($Mg^{2+}$) salt precipitations and catalyst poisoning[5,6]. Although seawater electrolysis in alkaline media can significantly reduce chloride evolution reaction (CER) selectivity and salt

precipitations, HER kinetics are over two orders of magnitude more sluggish than that in acidic electrolyte, resulting in increased overpotential and reduced energy conversion efficiency[7,8]. Considering the high cost and scarcity of Pt-based catalyst, the development of non-precious metal-based HER electrocatalysts with highly electrocatalytic activity and stability is indispensable for alkaline seawater electrolysis[9–11].

Transition-metal sulfides (TMSs) are among the most promising HER electrocatalysts due to their excellent electrocatalytic performance. These catalysts commonly undergo dynamic reconstruction of surface structure/composition during long-term reaction, yielding atomic

[1]Dalian National Laboratory for Clean Energy, Dalian Institute of Chemical Physics, Chinese Academy of Sciences, Dalian 116023, PR China. [2]University of Chinese Academy of Sciences, Beijing 100049, PR China. [3]CAS Key Laboratory of Science and Technology on Applied Catalysis, Dalian Institute of Chemical Physics, Chinese Academy of Sciences, Dalian 116023, PR China. ✉e-mail: byang@dicp.ac.cn; edwang@dicp.ac.cn

defects at interfaces[12,13]. Previous studies have highlighted that sulfur vacancies (Vs) can enhance catalytic activity to a certain extent. For example, Zhang et al. observed the structural evolution of multivalent nickel-base sulfides from $NiS_2$ to α-NiS, β-NiS and $Ni_3S_4$, where the diffusion and accumulation of Vs promote the HER performance[14]. Hu et al. similarly noted surface reconstruction in $(NiCo)S_{1.33}$ catalyst through sulfur atoms substitution with oxygen during reaction, thereby improving electrochemical reactivity[15]. While mild sulfur leching can promote electrochemical performance to some extent by forming active Vs sites or a reconstructed surface, long-term leching however, will consequently lead to unsatisfactory stability due to the loss of abundant sulfur components and rapid activity degradation. Particularly under alkaline solution, leaching emerges as a primary degradation mechanism for HER electrocatalysts due to differences in ion concentration between alkaline solution and pure-phase catalyst[16]. Therefore, achieving precise control of sulfur leaching in TMSs is crucial to maintaining high activity and durability towards high-performance HER.

Herein, we propose a strategy to engineer interfacial sulfur migration by constructing a CN@NiCoS heterostructure electrocatalyst with nitrogen doped carbon (CN) shell encapsulation. The NiCoS heterostructure stimulates sulfur migration, forming sulfur vacancies that facilitate rapid electron transfer for enhanced HER activity. Additionally, the CN overlayers encapsulation effectively confines leached sulfur atoms via strong C-S bonds at the interface, preventing sulfide dissolution into alkaline electrolyte and ensuring high durability. The catalytic performance of CN@NiCoS demonstrates low overpotentials of 4.6 and 8 mV at 10 mA cm⁻² in both alkaline freshwater and seawater media, along with long-term stability up to 1000 h, surpassing the reported TM-based electrocatalysts in literature. State-of-the-art ex situ/in situ characterizations further reveal the dynamic formation of S-doped CN network and sulfur vacancies pairing sites (S/NC@NiCoS-Vs) at the interface that are responsible for the enhanced activity and stability of CN@NiCoS during the HER reaction.

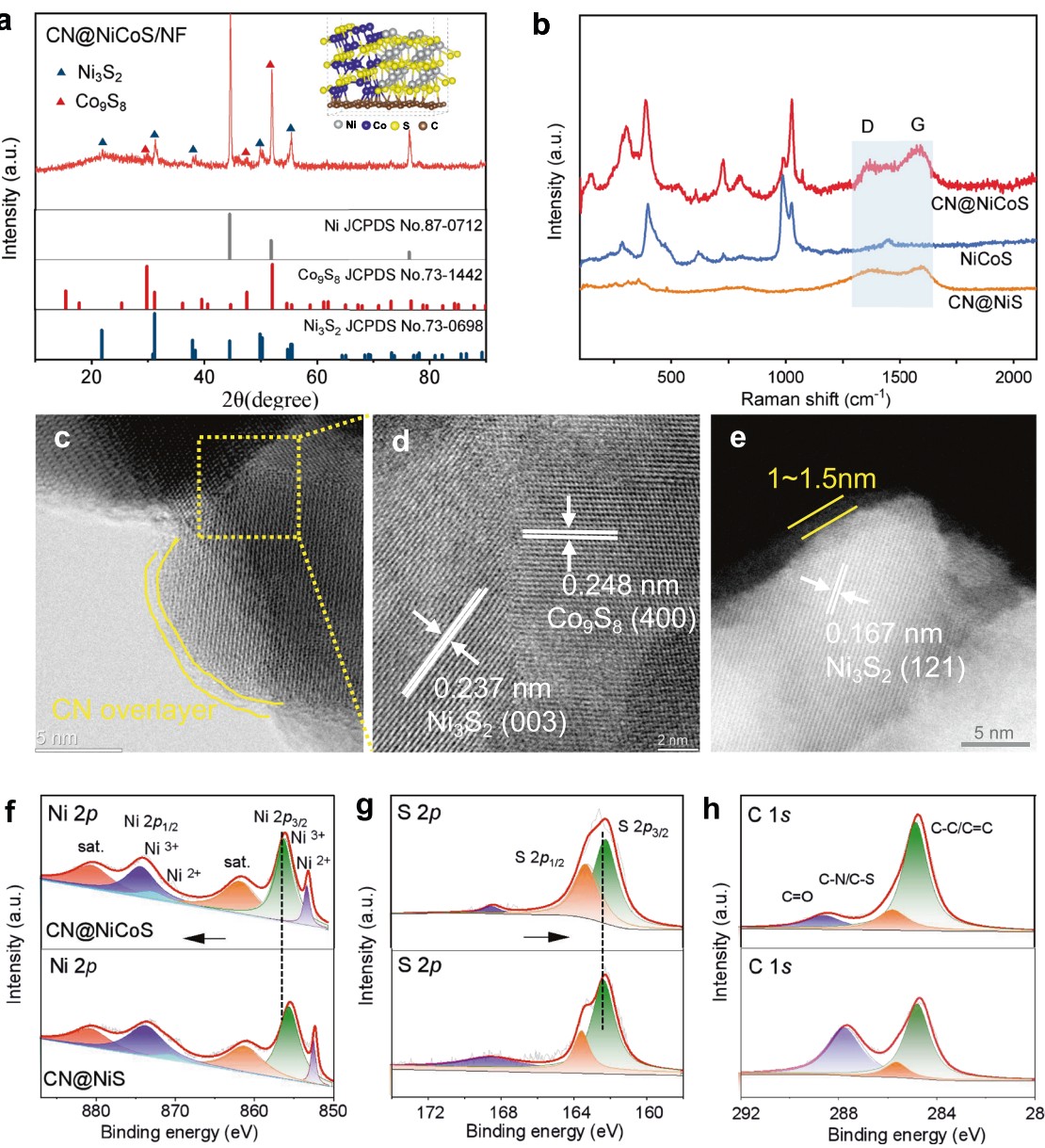

**Fig. 1 | Structural characterization of CN@NiCoS heterostructure electrocatalyst. a** XRD profile of CN@NiCoS/NF. **b** Raman spectra of NiCoS (blue), CN@NiS (orange) and CN@NiCoS (red). **c–e** HRTEM image of CN@NiCoS (**d** is the enlarged area of the image in **c**). High-resolution XPS spectra of CN@NiS and CN@NiCoS (**f**) Ni 2*p*, (**g**) S 2*p*, (**h**) C 1*s*.

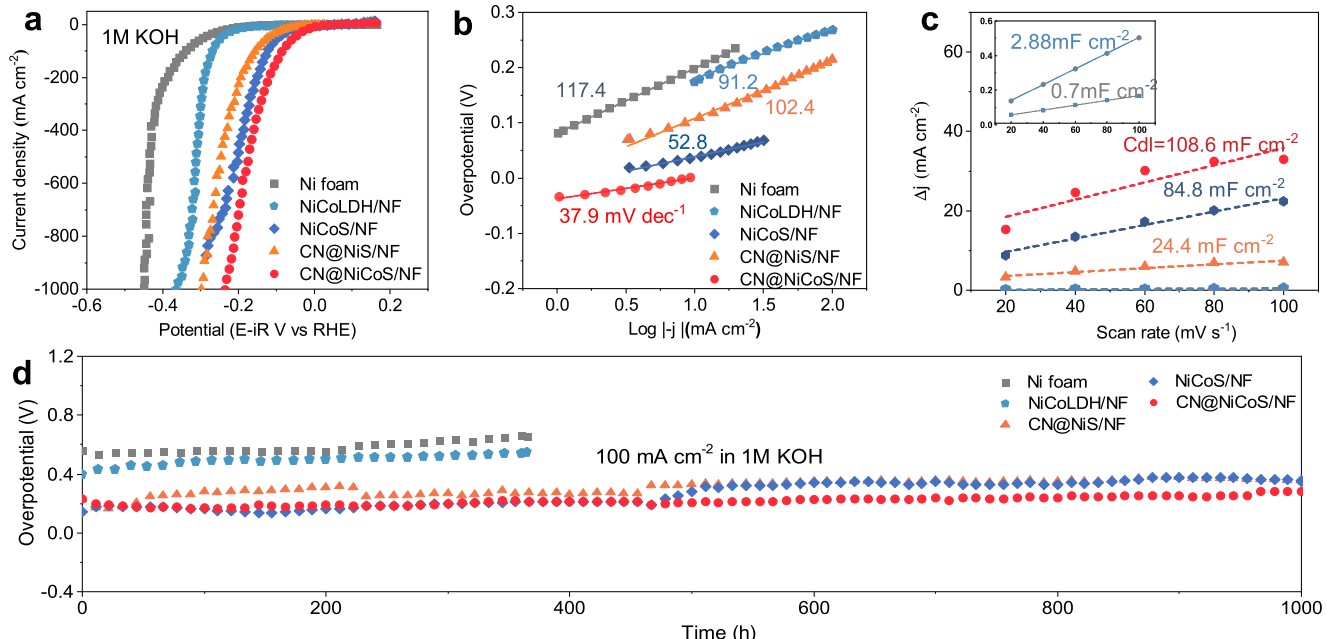

**Fig. 2 | Electrochemical performance of CN@NiCoS catalyst in alkaline water.** HER performances of Ni foam (gray), NiCoLDH/NF (green), NiCoS/NF (blue), CN@NiS/NF (orange) and CN@NiCoS/NF (red) electrodes (electrode area: 1 cm × 1 cm) in 1 M KOH solution. **a** LSV curves with iR-corrected. **b** Tafel plots. **c** Plots of current density difference against scan rates. **d** Galvanostatic measurement at $j = -100\,mA\,cm^{-2}$.

## Results

### Structural identification of CN@NiCoS heterostructure electrocatalyst

CN@NiCoS was synthesized via hydrothermal-sulfidation/carbon-coating process, while CN@NiS, CN@CoS and NiCoS were prepared for comparison using a similar method (see Methods section). The X-ray diffraction (XRD) pattern (Fig. 1a) of CN@NiCoS on Ni foam (NF) electrode shows characteristic diffraction peaks of $Ni_3S_2$ (JCPDS No. 73-0698) and $Co_9S_8$ (JCPDS No. 73-1442), along with diffractions signals from the Ni foam substrate. Corresponding Raman spectra (Fig. 1b) of the CN@NiS and CN@NiCoS further confirm the presence of CN overlayers with distinct D and G bands at 1367.8 and 1593.4 $cm^{-1}$ [17], respectively. The absence of prominent carbon diffraction in XRD thus reflects its amorphous nature and ultrathin thickness. Transmission electron microscopy (TEM) and high-angle annular dark-field scanning transmission electron microscopy (HAADF-STEM) images of CN@NiCoS (Fig. 1c) depict an ultrathin CN overlayer approximately 1~1.5 nm encapsulating the outer surface of the NiCoS particle. High-resolution TEM (HRTEM) images (Fig. 1d) clearly resolve a $Ni_3S_2/Co_9S_8$ heterostructure within the NiCoS nanoparticle, with interplanar spacing of 0.237 and 0.248 nm corresponding to $Ni_3S_2$ (003) and $Co_9S_8$ (400) planes. These results collectively elucidate the unique structure of the CN@NiCoS catalyst, featuring an ultrathin CN shell encapsulating $Ni_3S_2/Co_9S_8$ heterostructure, which is distinct notably from CN@NiS (single $Ni_3S_2$ phase), CN@CoS (single $Co_9S_8$ phase) and NiCoS (without the C layer), as shown in Fig. 1e and Supplementary Fig. 1 and 3).

The surface compositions and electronic structures of CN@NiCoS and CN@NiS catalysts were investigated further using X-ray photoelectron spectroscopy (XPS), as depicted in Fig. 1f–h. In the Ni 2P XPS spectrum of pristine CN@NiCoS (Fig. 1f), distinct $Ni^{2+}$ and $Ni^{3+}$ components are clearly identified with characteristic Ni $2p_{3/2}$ peak at 855.9 and 861.7 eV, respectively, indicative of a typical $Ni_3S_2$ phase[18,19]. Moreover, the Co 2p XPS profiles (Supplementary Fig. 2a) show deconvoluted peaks at 797.6 and 781 eV, accompanied by satellite peaks at 778.7 and 795.3 eV, demonstrate the presence of the $Co^{2+}$ and $Co^{3+}$ species in $Co_9S_8$ phase[20,21]. The characteristic peaks of S $2p_{3/2}$ and S $2p_{1/2}$ located at 161.7 and 162.8 eV can be ascribed to metal sulfur (M-S) bonds (Fig. 1g)[22,23] in the $Ni_3S_2$ and $Co_9S_8$ phase. Notably, compared to CN@NiS, the upshift of Ni 2p along with the opposite downshift of S 2p in CN@NiCoS strongly infers electron transfer from Ni sites to adjacent S atoms, likely due to the formation of $Ni_3S_2/Co_9S_8$ heterojunction as observed in Fig. 1c and Supplementary 1c. The charge distribution at the $Ni_3S_2/Co_9S_8$ interface can significantly promote HER performance due to fast electron transfer[24]. The C 1s spectra (Fig. 1h) reveals three distinct carbon species at 284.7, 285.7 and 288.5 eV, attributed to C-C/C=C, C-N/C-S and C=O species, respectively[25]. Furthermore, intensified N 1s signals in the CN@NiCoS (Supplementary Fig. 2b)) further demonstrate the presence of doped nitrogen in the CN overlayer.

### HER performance in alkaline water and seawater

The HER performance of various TM catalysts (Ni foam, NiCoLDH, NiCoS, CN@NiS, CN@CoS and CN@NiCoS) was evaluated using linear scan voltammogram (LSV) in 1 M KOH alkaline media. As shown in Fig. 2a and Supplementary Fig. 4a, the CN@NiCoS catalyst exhibits a remarkably low overpotential of only 4.6 mV to deliver a current density of 10 mA $cm^{-2}$, which is far lower than that of NiCoS (37 mV), CN@CoS (58.9 mV), CN@NiS (59 mV), NiCoLDH (173.9 mV) and NF (198.8 mV). Furthermore, the CN@NiCoS catalyst also delivers low overpotential at higher current density (83.9 mV at $\eta_{100}$, 173.6 mV at $\eta_{500}$ and 236 mV at $\eta_{1000}$). Tafel slope analysis (Fig. 2b and Supplementary Fig. 4b) shows that the CN@NiCoS catalyst achieves a lower Tafel slope (37.9 mV $dec^{-1}$) compared to NiCoS (52.8 mV $dec^{-1}$), CN@CoS (53.8 mV $dec^{-1}$), CN@NiS (102.4 mV $dec^{-1}$), NiCoLDH (91.2 mV $dec^{-1}$) and NF (117.4 mV $dec^{-1}$), indicating the dominant Volmer-Heyrovsky mechanism in the HER process.

To further elucidate the electron transfer dynamics, electrochemical impedance spectra (EIS) measurement was conducted at stationary potential (Supplementary Figs. 6a, 4c). The CN@NiCoS catalyst exhibits the smallest charge transfer resistance ($R_{ct}$) among all studied catalysts, implying faster charge transfer kinetics at the

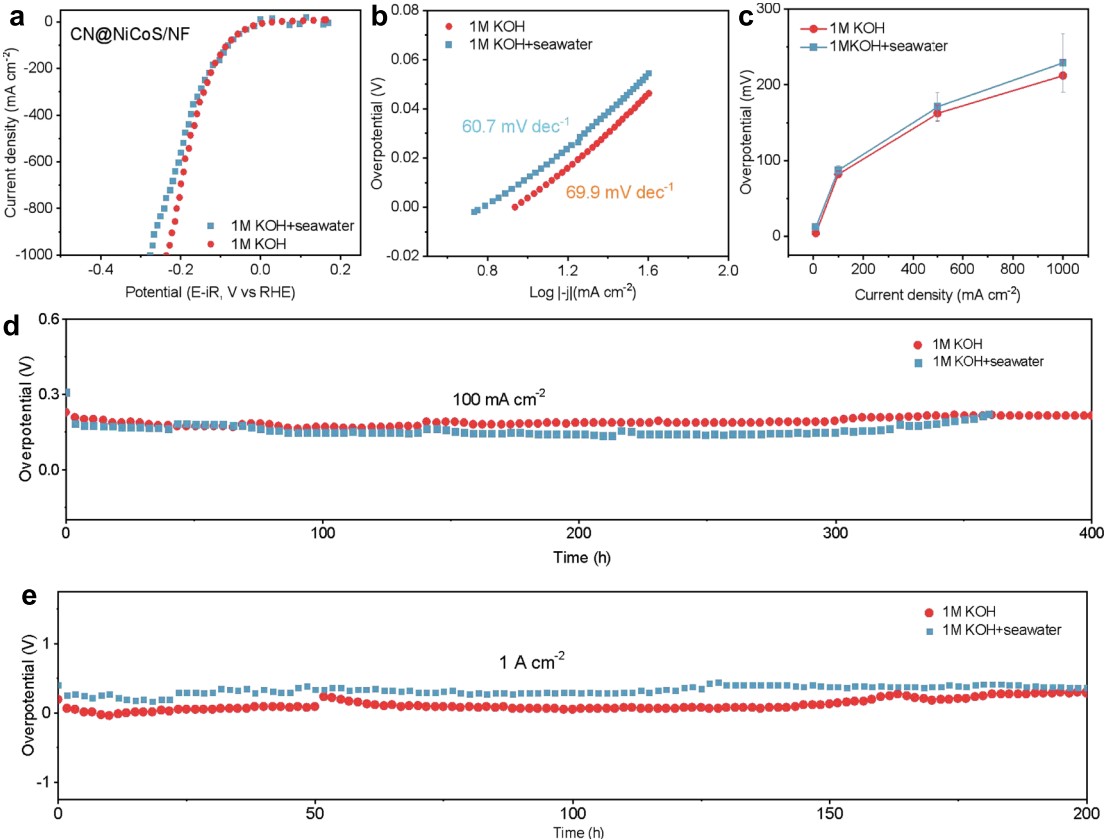

**Fig. 3 | Electrochemical performance of CN@NiCoS catalyst in seawater.** HER activity in 1 M KOH (red) and 1 M KOH + seawater (green). **a** HER polarization curves. **b** Tafel plots. **c** HER overpotentials at $j$ = 10, 100, 500 and 1000 mA cm$^{-2}$, respectively. Durability test in different electrolyte in 100 mA cm$^{-2}$ (**d**) and in 1 A cm$^{-2}$ (**e**).

catalyst/electrolyte interface. The electrochemical activation surface area (ECSA) correlated linearly with the double-layer capacitance $C_{dl}$ (Fig. 2c and Supplementary Fig. 4d). The CN@NiCoS catalyst exhibits a larger $C_{dl}$ value (108.6 mF cm$^{-2}$) compared to NiCoS (84.8 mF cm$^{-2}$), CN@NiS (24.2 mF cm$^{-2}$), CN@CoS (10.6 mF cm$^{-2}$), NiCoLDH (2.88 mF cm$^{-2}$) and NF (0.7 mF cm$^{-2}$), unveiling exposure of more electrochemically active sites. To reveal the intrinsic HER activities, the activation energy ($E$a) of studied electrocatalysts were compared (Supplementary Fig. 6b). According to Arrhenius equation, the $E$a of CN@NiCoS (5.7 kJ mol$^{-1}$) is lower compared to NiCoS (28.97 kJ mol$^{-1}$), CN@NiS (21.12 kJ mol$^{-1}$), NiCoLDH (36.4 kJ mol$^{-1}$) and NF (11.88 kJ mol$^{-1}$). Notably, the CN overlayer alone exhibits almost no HER activity (Supplementary Fig. 7). The enhancement of HER activity of NiCoS due to CN encapsulation strongly suggests the promotional effect of CN overlayer that modulates the electronic structure of NiCoS heterojunction for enhanced HER activity.

Long-term stability is a prerequisite for the feasibility of practical application. We evaluated different electrocatalysts at a high current density of 100 mA cm$^{-2}$ (Fig. 2d). Apparently, Ni foam and NiCoLDH require significantly higher overpotentials to maintain a constant current density. In comparison, CN@NiS exhibits a stable smooth polarization curve but with higher overpotential, while NiCoS catalyst initially shows low potential but rapidly deteriorates, resulting in unsatisfactory HER stability. Remarkably, the CN@NiCoS catalyst achieves long-term stability at 100 mA cm$^{-2}$ over 1000 h in 1 M KOH electrolyte, with no distinct degradation and lower overpotential compared to other reported catalysts in the literatures[7,24,26–49], by comparing activity (e.g. A-CoB/Mxene (($\eta_{10}$ = 15 mV)[48], Ni$_3$Sn$_2$-NiSn$_2$O$_x$ ($\eta_{10}$ = 14 mV)[28], and stability (e.g. GDY/MoO$_3$ (120 h; 100 mA cm$^{-2}$)[37], MoC-Mo$_2$C (1000 h; 30 mA cm$^{-2}$)[38] and C-Co-MoS$_2$ (240 h; 100 mA cm$^{-2}$)[45]) (Supplementary Fig. 8 and Tables 1, 2). The existence

of CN layer improves catalyst stability, while the Ni$_3$S$_2$/Co$_9$S$_8$ heterostructure delivers high intrinsic activity through rapid electron transfer, which breaks the activity-stability trade-offs of conventional TMSs electrocatalysts.

Furthermore, we systematically assessed the HER activity of CN@NiCoS catalyst in alkaline seawater electrolyte to demonstrate its excellent durability against the impurity ion. LSV curves (Fig. 3a) reveal that the CN@NiCoS requires only 8 mV overpotential to deliver 10 mA cm$^{-2}$ current density in alkaline seawater media. Additionally, the CN@NiCoS catalyst exhibits a considerably low Tafel slope of 39.4 mV dec$^{-1}$ (1 M KOH) and 68.9 mV dec$^{-1}$ (1 M KOH + seawater), as shown in Fig. 3b. Notably, even at high current densities of 100, 500 and 1000 mA cm$^{-2}$, CN@NiCoS maintains low overpotential of 79.8, 193.2 and 281 mV, respectively, in the corresponding alkaline seawater electrolyte (Fig. 3c), highlighting the promotional role of bifunctional Ni$_3$S$_2$/Co$_9$S$_8$ interface and CN overlayer in improving the HER activity of alkaline seawater. We observed that the intrinsic HER activity in alkaline seawater has an attenuated tendency compared to that in 1 M KOH electrolyte, likely due to the ion impurities[50]. Meanwhile, the CN@NiCoS catalyst also presents excellent stability in alkaline seawater, maintaining performance for over 400 and 200 h at 100 and 1000 mA cm$^{-2}$ (Fig. 3d, e), reflecting the protective role of CN overlayers against electrolyte impurities and the great potentiality of CN@NiCoS for industrial seawater electrolysis. Supplementary Fig. 9 further demonstrates the reproducibility of the CN@NiCoS catalyst, in either 1 M KOH or 1 M KOH + seawater solutions. To further demonstrate the potential of the CN@NiCoS catalyst in practical operating conditions, we evaluated its electrochemical performance under various solvent environments and pH conditions (Supplementary Figs. 10, 11)[51–53], and achieved the optimal HER activity in a 1 M KOH solution.

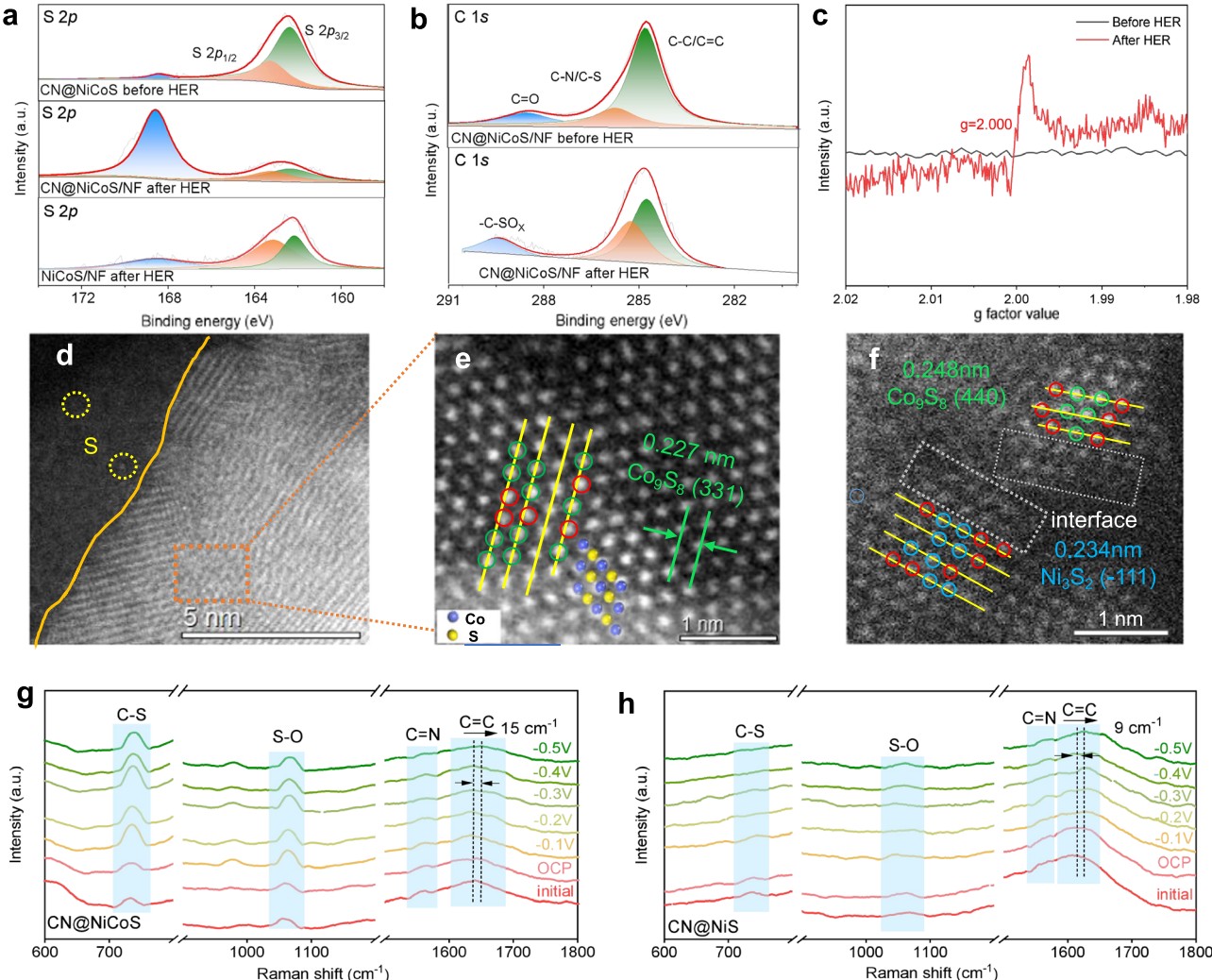

**Fig. 4 | Identification of Sulfur migration at CN@NiCoS interface.** S 2*p* (**a**) and C 1*s* (**b**) XPS spectra of CN@NiCoS and NiCoS before and after 100 h stability test in 1 M KOH solution. **c** The electron paramagnetic resonance (EPR) of CN@NiCoS before and after HER stability. HAADF-STEM images of CN@NiCoS (**d**–**f**) after HER stability test. The green, yellow and red circles represent Co, S atoms and S vacancy respectively. In situ Raman spectra of CN@NiCoS (**g**) and CN@NiS (**h**) at different potential.

## Dynamic sulfur migration of CN@NiCoS under working condition

To unveil the surface reconstruction of catalytic active sites, state-of-the-art ex situ/in situ characterizations were employed to monitor the structure evolution of CN@NiCoS catalysts during the HER process. According to XRD and TEM results (Supplementary Fig. 12a, b), the bulk phase composition of the CN@NiCoS catalyst remain unchanged before and after HER process, suggesting the structural stability of heterojunction during reaction. This stability can be further supported by XPS survey spectra (Supplementary Fig. 12c), without noticeable changes in the Ni 2*p*, Co 2*p*, N 1*s* and C 1*s* spectra, while the S 2*p* spectrum shows a significant enhancement after HER process. As shown in Fig. 4a, unlike the pristine CN@NiCoS, the intensity of the S-O peaks at 168.8 eV increases significantly after HER, indicating more S incorporation into the electrode surface due to enhanced surface sensitivity of XPS techniques. The increased peak intensities of -C-SO$_x$ and C-N/C-S peaks in the C 1*s* spectrum after HER (Fig. 4b) further validate the migration of S atoms from the Ni$_3$S$_2$/Co$_9$S$_8$ heterojunction to the CN overlayer, forming a C-S bond[54] during HER process. Meanwhile, the Ni 2*p*$_{3/2}$ and Co 2*p*$_{3/2}$ peaks of the CN@NiCoS catalyst post-HER (Supplementary Fig. 13) also shift to lower binding energy (BE) of 855.7 eV and 782.3 eV respectively, while the S 2*p*$_{3/2}$ signals

(Fig. 4a) distinctly shift towards a higher BE of 161.8 eV. These opposite shifts can be ascribed to the reduction of metal centers and the formation of S vacancies (Vs) on CN@NiCoS during the outbound migration of S atoms into the CN overlayers[55]. The generation of Vs on CN@NiCoS was further verified by the electron paramagnetic resonance (EPR) spectra (Fig. 4c), showing an enhanced EPR signal at $g = 2.000$ after HER process[56,57]. Similarly, the HAADF-STEM image of CN@NiCoS after HER (Fig. 4d) also reveals trapped S atoms on the outer CN overlayer, evident from enhanced Z-contrast[57,58]. The atomic arrangement of Co$_9$S$_8$ from the view of the (331) plane is illustrated in Fig. 4e. The green circles emphasize the crystal surface of Co$_9$S$_8$ (331) without deformation, while the red circles display the crystal surface of Co$_9$S$_8$ (331) is slightly deformed, as a characteristic feature indued by S vacancies in TMSs[59]. Additionally, the Ni$_3$S$_2$ (−111) also exhibits noticeable deformation with similar atomic rearrangement. Especially at the Ni$_3$S$_2$/Co$_9$S$_8$ interfacial region, the dim features of defected sites/missing atoms are clearly resolved as highlighted with dashed boxes, suggesting highly defective Ni$_3$S$_2$/Co$_9$S$_8$ interface (Fig. 4f and Supplementary Fig. 14) compared to the fresh sample before HER (Fig. 1d). This reflects the preferential formation of S vacancies at the Ni$_3$S$_2$/Co$_9$S$_8$ interface, leading to a more disordered lattice and missing atoms (dim features) at the interface[60]. Collectively, these findings

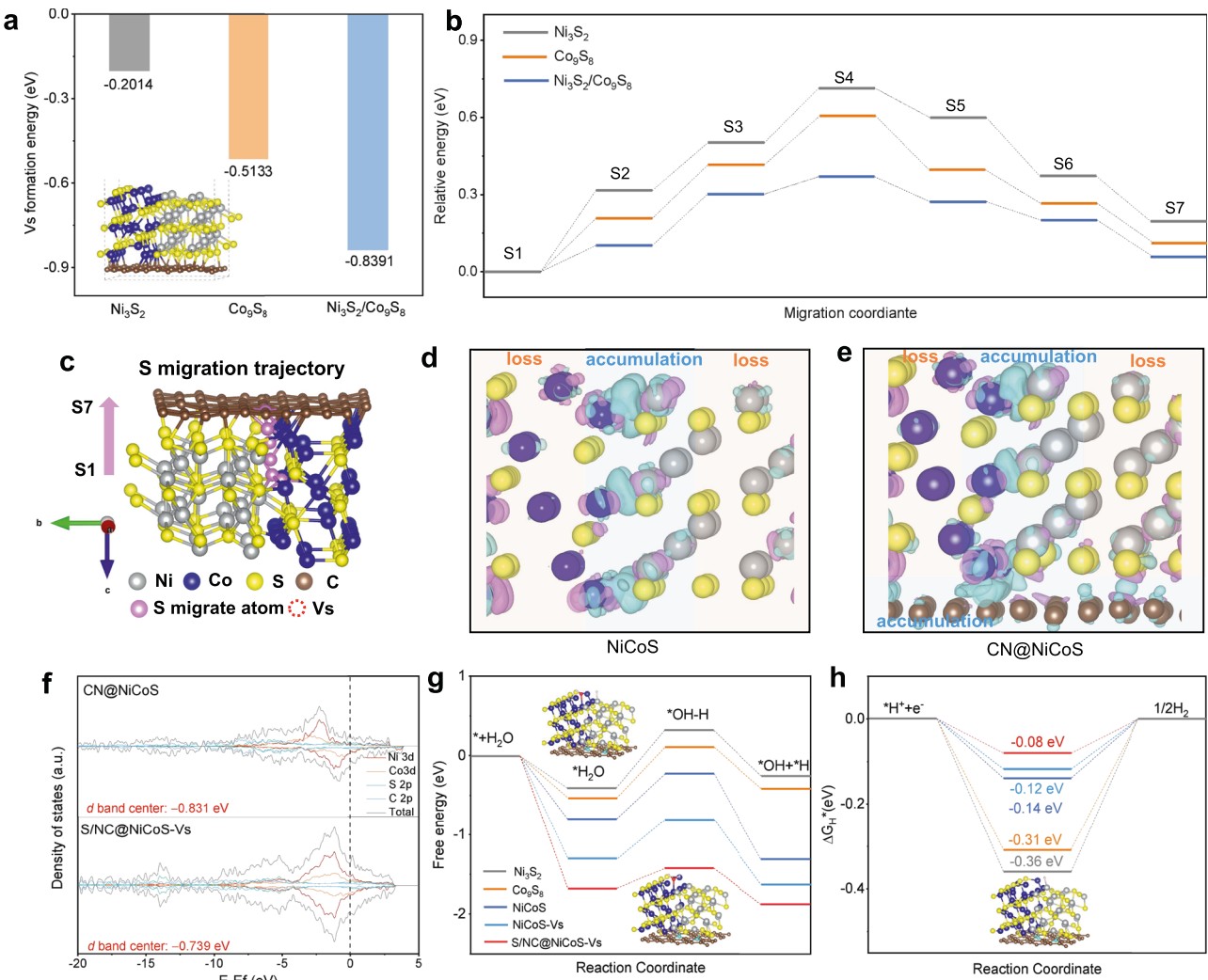

**Fig. 5 | Density Functional Theory (DFT) calculations for CN@NiCoS hetero-structure. a** The formation energy of S vacancy for $Ni_3S_2$, $Co_9S_8$, NiCoS hetero-juction. **b** Energy barrier of sulfur migration for CN@NiCoS heterostructure. **c** Corresponding sulfur migration trajectory in CN@NiCoS. Charge density difference of NiCoS (**d**) and CN@NiCoS (**e**). The isosurface value is 0.01 e/ $Å^3$, where green and red contours represent the electron accumulation and loss, respectively. **f** The density of states (DOS) of CN@NiCoS and S/NC@NiCoS-Vs. Reaction free-energy diagram of $Ni_3S_2$ (gray), $Co_9S_8$ (orange), NiCoS (blue), NiCoS-Vs (green) and S/NC@NiCoS-Vs (red), for Water dissociation (**g**), and Hydrogen evolution (**h**).

provide compelling evidence for the migration of S atoms from the NiCoS surface to encapsulated CN overlayers, leaving S vacancies at the NiCoS heterointerface and S dopants in the CN overlayers through the forming covalent C-S bonds.

To further verify the entrapment of sulfur within the CN layer, we measured the sulfur content in the electrolyte and on the surface of the NiCoS and CN@NiCoS catalyst after HER. The S 2p XPS spectrum (Fig. 4a) reveals that the CN@NiCoS exhibits a more distinct S-O component compared to NiCoS after HER, indicating a higher content of surface S species on the CN@NiCoS electrode due to CN encapsulation. Additionally, ICP-OES analysis further reveals a significant increase in S content in the electrolyte of NiCoS after HER, being eighteen times higher than that of CN@NiCoS (Supplementary Table 3), suggesting the irreversible sulfur leaching from NiCoS into the electrolyte during HER. This substantial loss of S atoms into the electrolyte may lead to severe collapse/deformation of the NiCoS structure after HER (Supplementary Fig. 15). Accordingly, we can thus conclude that the formation of $Ni_3S_2$/$Co_9S_8$ heterojunction in CN@NiCoS stimulates the S migration, which is subsequently captured by CN overlayers through the forming of C-S bonds. The as-formed

S/CN@NiCoS-Vs pairing sites promote efficient charge transfer at the heterojunction, thereby retaining S migration into the electrolyte and improving both activity and long-term stability of the catalyst. Conversely, the NiCoS catalyst without a carbon protection layer continuously loses sulfur into the electrolyte during reaction, eventually leading to deactivation.

The dynamic migration and trapping of S atoms on CN@NiCoS and CN@NiS during HER process were further observed by in situ Raman spectroscopy (experimental setup detailed in Supplementary Fig. 16). With the applied potential decreases to −0.5 V, the intensities of C-S and S = O bands of CN@NiCoS (Fig. 4g) at 739.4 and 1068.4 $cm^{-1}$ simultaneously intensify compared to the open circuit potential (OCP)[61]. In contrast, the weak C-S and S = O bands on CN@NiS (Fig. 4h) manifests that the $Ni_3S_2$/$Co_9S_8$ heterostructure in CN@NiCoS is more favorable to stimulating sulfur migration. The strong C-S interaction further implies the inhibition of sulfur leaching into the electrolyte due to the encapsulation of CN overlayers, that improves long-term stability. The C = C bands of CN@NiCoS also shows a significant redshift of 15 $cm^{-1}$, greater that of CN@NiS (9 $cm^{-1}$), suggesting weakened C = C vibrations due to sulfur incorporation into the CN overlayers.

Meanwhile, we also notice a slightly decreasing intensity of the C = N peak (Fig. 4g) along with the appearance of metal-N peak[62] in CN@NiCoS after HER (Supplementary Fig. 17), indicating the sulfur incorporation may weaken the C = N interaction and enhance metal-N interaction with electron deficient $Ni^{2+}/Co^{2+}$ sites, strengthening the interfacial structures of CN@NiCoS towards the long-term stability.

## Mechanism for stability and activity enhancement during long-term HER

To understand surface reconstruction induced by S migration and its effect on electrocatalytic activity, we calculated the formation energy of S vacancies and migration pathways with corresponding energy barrier in various sulfides. The formation energy of S vacancies in the $Ni_3S_2$/$Co_9S_8$ heterostructure (−0.84 eV) is more negative compared to $Ni_3S_2$ (−0.20 eV) and $Co_9S_8$ (−0.51 eV) phases, indicating that S vacancy formation in the NiCoS heterostructure is thermodynamically favored over $Ni_3S_2$ and $Co_9S_8$ phases alone[63,64]. This observation is consistent with our STEM analysis, showing defective $Ni_3S_2/Co_9S_8$ interface on CN@NiCoS after HER (Fig. 4f and Supplementary Fig. 14). To investigate the S migration mechanism, we explored multiple pathways and corresponding energy barrier for $Ni_3S_2$, $Co_9S_8$ and NiCoS (Fig. 5b, c). The $Ni_3S_2/Co_9S_8$ interface exhibits the lowest migration energy barrier, manifesting more favorable sulfur migration in the NiCoS heterojunction. As depicted in Fig. 5b, the process begins with the activation and cleavage of the metal-S bond (S2), followed by the adsorption of free S atoms at the metal site (S3). These adsorbed S atoms then occupy adjacent defect sites, creating an S vacancy at the original position (S4), likely representing a transition state with the highest energy and an imaginary frequency (Supplementary Fig. 18). Subsequently, these atoms migrate across NiCoS (S5 and S6) and are ultimately trapped by the CN shell through the formation of an S-C bond (S7). Throughout this process, we infer that lattice mismatch/rearrangement at the $Ni_3S_2$/$Co_9S_8$ interface favors S vacancies formation (Figs. 4d–f, 5a) and thus facilitates the S migration along the heterojunction.

Figure 5d, e illustrates the charge density difference, revealing electron accumulation along the NiCoS interface and CN overlayer. The electron enrichment along the NiCoS interface and CN overlayer can promote the H* and H2O* adsorption, serving as catalytically active sites for HER[42]. This can be observed in Fig. 5f–h and Supplementary Fig. 19 that the NiCoS heterostructure possess stronger H2O adsorption ability (−0.79 eV), H2O dissociation energy barrier (0.57 eV), optimal hydrogen adsorption free energy (−0.14 eV) and higher density of states (DOS) intensity compared to $Ni_3S_2$ and $Co_9S_8$. We further explore the impact of Vs and S doping on HER activity by DFT calculations. By incorporating S vacancies, the NiCoS-Vs possess lower energy barrier (Fig. 5g, h), higher $d$-band center tower Fermi level and fast reaction kinetics process (Supplementary Fig. 20) compared to NiCoS alone. This result indicates that the presence of Vs in NiCoS heterostructures can regulate the orbital distributions ($d$-band center) with higher electron states near Fermi level and introduce more coordinatively unsaturated sites[65,66], thereby adjusting H* adsorption and accelerating the HER kinetic process. Furthermore, the S and N doped C layer (S/N-C) displayed more optimized $\Delta G_{H2O}$ and $\Delta G_{H*}$ values compared to N-C (Supplementary Fig. 21e, f), which aligns with experimental observations that S/N-C displays rapid HER dynamic with lower overpotential, Tafel slope and charge transfer impedance (Supplementary Fig. 21a−c). This finding reveals that the formation of S-doped C after sulfur migration further improve hydrogen evolution activity due to the redistribution of the electronic structure.

The dynamic trapping of sulfur atoms by the CN overlayer not only ensures the long-term stability of catalyst, but also enhances its electrocatalytic activity. Comparing with pristine CN@NiCoS, the promotional role of S/NC@NiCoS-Vs was further elucidated by DFT calculations. Figure 5f illustrates that S/NC@NiCoS-Vs exhibits significantly higher DOS intensity and $d$-band center close to Fermi energy level than CN@NiCoS. Moreover, S/NC@NiCoS-Vs demonstrates stronger H2O adsorption (−1.68 eV), optimal H2O dissociation (0.26 eV) and H adsorption energy (−0.08 eV) compared to CN@NiCoS (Fig. 5g, h), indicating accelerated kinetics for water dissociation and hydrogen evolution facilitated by S-Vs pair formation. The role of S-Vs pair may be attributed to the creation of localized states by S vacancies, which induces a strong $d$-$d$ orbital overlap (Ni 3$d$ and Co 3$d$). Additionally, S-doped C activates additional $d$ electron states near the Fermi level, promoting charge transfer and H* adsorption for enhanced HER activity. Furthermore, the as-formed C-S bridge bond, thus providing pathways for more available electrons migration, to participate in HER reaction and achieve high activity and stability alkaline HER[67].

## Discussion

In summary, we designed a CN@NiCoS electrocatalyst with strikingly high intrinsic HER activity ($\eta_{10}$ = 4.6 mV; 8 mV) and long-term stability (100 mA cm$^{-2}$, 1000 h; 350 h) for alkaline freshwater and seawater. The dynamic evolution of interface structure and controlled sulfur migration was clearly demonstrated. The $Ni_3S_2$-$Co_9S_8$ heterostructure stimulates the formation of sulfur vacancies that facilitate fast electron transfer and thus accelerate water adsorption/dissociation in alkaline media for enhanced HER activity. The leached S atoms was encapsulated by CN overlayers, preventing the dissolution of sulfides into alkaline electrolyte by forming strong C-S bond. The dynamic formation of sulfur dopant-vacancy (S/NC@NiCoS-Vs) paring site strongly enhanced the HER activity by modulation of d-band center at Fermi energy level, and remarkably improve the durability via confined sulfur migration. Our results thus provide a strategy to break the activity-stability trade-offs in TMSs by regulation of sulfur migration at the interfaces toward high-performance electrocatalyst for hydrogen production from seawater. The inhibition of sulfur leaching will also benefit the practical implementation of environmentally friendly TMSs catalysts for large-scale seawater electrocatalysis.

## Methods

### Materials

Commercial Ni foam (NF) was purchased from Tianyu New energy Technology Co., Ltd. Cobalt nitrate hexahydrate (Co(NO3)2·6H2O, 99%), urea (CH4N2O, 99%), ammonium fluoride (NH4F, 96.5%), thiourea (CH4N2S, 99%), potassium hydroxide (KOH, 85%) and hydrochloric acid (HCl, 36 - 38%) were purchased from Aladdin Reagent Co., Ltd. All the chemicals were used directly without further purification.

### Pretreatment of Ni foam

Firstly, commercial NF was ultrasonically washed with 1 M HCl solution, acetone and deionized (DI) water for 10 min, respectively. And then vacuum dried at 60 °C. Secondly, NiCoLDH was hierarchical grown on NF surface through hydrothermal treatment. Typically, 0.1 mmol Co(NO3)2·6H2O, 0.1 mmol urea and 0.3 mmol NH4F were dissolved in 60 ml DI water and stirred for 30 min to obtain uniform solution.

### Preparation of NiCoLDH/NF electrode

The pretreated NF was immersed into above solution and transferred to 100 ml Teflon-lined autoclave, which was sealed and maintained at 120 °C for 10 h. Finally, the obtained electrode was cleaned with ethanol and DI water for several times, followed by drying at 60 °C. The obtained electrode was denoted as NiCoLDH/NF (Ni: 6.05 wt%, Co: 8.38 wt%, obtained from X-ray photoelectron spectroscopy, XPS).

### Preparation of CN@NiCoS/NF electrode

The as-prepared NiCoLDH/NF electrode and CH4N2S powders were placed in separated ceramic boat, which were annealed in N2 atmosphere at 350 °C for 2 h with a heating rate of 2 °C min$^{-1}$. For contrast, the CN@NiS/NF electrode was acquired without the addition of Co(NO3)2·6H2O. The NiCoS/NF electrode was prepared by using sulfur

powder instead of thiourea. The obtained electrode was denoted as CN@NiCoS/NF (Ni: 3.65 wt%, Co: 0.71 wt%, S: 6.56 wt%, obtained from XPS).

## Materials characterization

The microstructure was analyzed by Transmission electron microscopy (TEM, JEM2100) and Scanning electron microscope (SEM, JSM-7800F). Aberration-corrected HAADF-STEM images were acquired on a JEM-ARM200F (JEOL), which is a thermal-field emission microscope with a probe Cs corrector, and the work voltage is 200 kV. For the HAADF imaging, a convergence angle of ~23 mrad and collection angle range of 68–174 mrad were adapted for the enhanced $Z$-contrast. X-ray photoelectron spectroscopy (XPS) was performed on a Thermo Scientific K-Alpha equipped with an Al Kα X-ray source (1486.6 eV). All spectra were acquired at room temperature and calibrated by setting the C $1s$ peak to 284.8 eV. X-ray diffraction (XRD) patterns was acquired using a Rigaku MiniFlex 600 with CuKα radiation ($\lambda$ = 1.54056 Å) from 6 to 90° at a rate of 5°min$^{-1}$. The Electron paramagnetic resonance (EPR) was carried out on a Bruker A200. Inductively coupled plasma-optical emission spectrometry (ICP-OES) was recorded on Perkin Elmer 7300DV.

## In situ Raman measurement

In situ Raman spectra were acquired on NanoWizard equipped with a laser excitation source of 532 nm, and Raman peaks were calibrated with a Si wafer before testing. The in situ three electrode electrochemical cell was conducted by CHI660E electrochemical workstation, which used CN@NiCoS/CN@NiS, Ag/AgCl and Pt wire as working electrode, reference electrode and counter electrode, respectively. In situ Raman spectra were performed at different applied potential from 0 to −0.5 V vs RHE (with an interval of 0.1 V).

## Electrochemical measurements

Electrochemical tests were carried out at 25 °C via electrochemical workstation (1287 electrochemical interface and 1260 A impedance analyzer, Solartron Analytical) with a typical three-electrode system. The NF, NiCoLDH, NiCoS, CN@NiS and CN@NiCoS electrodes, saturated calomel electrode and NF electrode were used as working, reference and counter electrodes, respectively. The electrochemical tests were conducted in a typical H-type electrolytic cell with electrode area of 1 × 1cm$^2$ (catalyst loading of ~15 mg cm$^{-2}$ expect for NF substate). 30 ml electrolyte of 1 M KOH or 1 M KOH+seawater in cathode/anode chambers. 1 M KOH + seawater electrolyte is prepared by dissolving solid KOH in natural seawater (including ~0.5 M NaCl), followed by 5 h to settle sediment and adjusted pH to 14 (± 0.2). Afterward, the upper liquid is transferred to a sealed container. The linear sweep voltammetry (LSV) curves were recorded with the potential range of −0.1 - 0.7 V (vs. RHE) at a scan rate of 2 mV s$^{-1}$. The chronopotentiometry teat was collected at a current density of 100 mA cm$^{-2}$ to evaluate long-term stability. The electrochemical impedance spectroscopy (EIS) plots were observed at a −0.1 V vs. RHE potential from 10$^5$ to 1 Hz with a 5 mV amplitude. The double layer capacitance ($C_{dl}$) was estimated by cyclic voltammetry (CV) curves at scanning rates of 20, 40, 60, 80, and 100 mV/s. Tafel slopes were identified by LSV curves based on:

$$\eta = a + b\,log\,j \tag{1}$$

where $\eta$, j, a and b are the overpotential, current density and constant, respectively. All reported potentials have been converted to reversible hydrogen electrode (RHE) based Nernst equation:

$$E_{RHE} = E_{SCE} + 0.242 + 0.0591 \times pH \tag{2}$$

where $E_{RHE}$ and $E_{SCE}$ are the potential based on reversible hydrogen electrode and saturated calomel electrode, and pH is electrolyte value.

## DFT calculations

Due to the CN@NiCoS model's large size (approximately 160 atoms) and its complex triple-phase interfaces, we adopted the ultrasoft pseudo-potential for our calculations, ensuring improved computational efficiency while maintaining adequate precision. Density functional theory (DFT) calculations[68,69] was performed within the generalized gradient approximation (GGA) using the Perdew-Burke-Ernzerhof (PBE)[70] formulation. Projected augmented wave (PAW) potentials[71,72] was adopted using a plane wave basis set with a kinetic energy cutoff of 520 eV. The structural model for NiCoS heterostructure was constructed using Ni$_3$S$_2$ (−110) and Co$_9$S$_8$ (331) facets according to experimental observation (Supplementary Fig. 1). During structural optimizations, a 1 × 1 × 1 **k**-point grid was utilized for sampling the Brillouin zone. The bottom two atomic layers were held fixed while the remaining layers were allowed to relax. Structural relaxations were conducted until the total forces per atom were reduced to less than 10$^{-4}$ eV/Å, while ensuring the deviation from the target pressure < 0.1 GPa. Partial occupancies of the Kohn−Sham orbitals were allowed using the Gaussian smearing method (width of 0.02 eV). The electronic energy was considered self-consistent when the energy change was smaller than 10$^{-5}$ eV. A geometry optimization was considered convergent when the energy change was smaller than 0.03 eV Å$^{-1}$. The vacuum spacing is 20 Å for the slab. Brillouin zone integration is performed using 2 × 2 × 1 Monkhorst-Pack **k**-point sampling for DOS calculations.

The adsorption energies ($E_{ads}$) were calculated as:

$$E_{abs} = E_{ad/sub} - E_{ad} - E_{sub} \tag{3}$$

where $E_{ad/sub}$, $E_{ad}$, and $E_{sub}$ are the total energies of the optimized adsorbate/substrate system, the adsorbate in the structure, and the clean substrate, respectively.

The free energy was calculated using the equation:

$$\Delta G = E_{ads} + ZPE - TS \tag{4}$$

where $\Delta G$, $E_{ads}$, $ZPE$ and TS are the free energy, total energy from DFT calculations, zero-point energy and entropic contributions, respectively.

Ion migration energy was located by the nudged elastic band (NEB) method[73]. In the NEB method, the path between the initial and final state was discretized into a series of intermediate states. The corresponding optimized geometries were relaxed until the perpendicular forces were smaller than 0.05 eV/Å.

## Data availability

The computational model containing all atomic coordinates data in this study have been deposited in the Figshare database under accession code (https://doi.org/10.6084/m9.figshare.25772682). Additional data generated in this study are provided in the Supplementary Information/Source Data file. Source data are provided with this paper.

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

## Acknowledgements

This work was supported by National Key R&D Program of China (No. 2023YFE0125500, E.W), National Natural Science Foundation of China (No. 22072151, B.Y.; No. 22005299, Z.Y), the NSFC Center for Single-Atom Catalysis (No. 22388102, B.Y.), CAS Project for Young Scientists in Basic Research (YSBR-022, B.Y.), the Strategic Priority Research Program of the Chinese Academy of Sciences (Grant No. XDA22010601, E.W), the DNL Cooperation Fund, CAS (DNL202011, E.W).

## Author contributions

B.Y. and E.W. conceived and supervised the entire project. M.L. prepared the catalyst and conducted the electrochemical tests and basic characterizations. H.L. performed the STEM imaging and data analysis. M.L. and H.L. co-wrote the manuscript. B.Y. and E.W. contributed to revise this manuscript. H.F., Q.L., Z.Y., and A.W. participated in the data discussion and manuscript preparation.

## Competing interests

The authors declare no competing interests.
