## [Peer Review File · Nature Communications]

REVIEWER COMMENTS

Reviewer #1 (Remarks to the Author):

Though seawater as HER feedstock has attracted considerable attention, the hydrogen production from seawater is still challenging limited by the deactivation of HER electrode owing to chlorine corrosion, salts precipitations and catalyst poisoning. This manuscript submitted by Wang et al. constructed the CN@NiCoS heterostructure electrocatalyst with nitrogen doped carbon (CN) shell encapsulation to stimulate sulfur migration and facilitate electron transfer for enhanced HER activity. Moreover, they proved that the migrated S atoms were subsequently captured by the CN shell via strong C-S bond, so that to prevent the dissolution of sulfides into alkaline electrolyte and improve the stability of catalyst in alkaline freshwater and seawater media. This work provide a new guidance to control of the sulfur leaching in TMSs to maintain high activity and durability towards high-performance HER. Therefore, I recommend publishing this manuscript after minor revisions. Below is a list of specific comments and questions the authors should address before publication.

1. According to the HAADF-STEM image of CN@NiCoS after HER (Fig. 4e), there are obvious atomic arrangement in Co₉S₈. So, does Ni₃S₂ undergo a similar atomic rearrangement in this process? The authors have always emphasized that the heterogeneous interface of NiCoS promotes the migration of S, so how about the atomic arrangement at the interface before and after the HER?
2. In fact, the interaction of the interface between Ni₃S₂ and Co₉S₈ is not the same as the Co incorporation into Ni₃S₂ phase (line 131, page 6 and line 29, page 12), please seriously consider the difference between the interface and doping and make corresponding modifications.
3. It is suggested that the authors supplement the structural characterization and electrochemical data of CN@Co₉S₈ to further confirm the role of the heterogeneous interface.
4. There are some vague description and typo errors in this article. Thus, for a better readability, please check throughout the whole article text and correct all the errors.

Reviewer #2 (Remarks to the Author):

The manuscript introduces a new strategy employing a CN@NiCoS heterostructure electrocatalyst with a nitrogen-doped carbon shell encapsulation, demonstrating remarkable HER performance. The topic is interesting, however, the exploration of the theoretical mechanism appears immature.

1. The dynamics of sulfur (S) migration seemingly exert a pivotal influence on HER performance. It is imperative to provide a more profound insight into S migration, encompassing the migration pathways and energy barriers associated with S migration within the CN@NiCoS heterostructure. A comprehensive analysis elucidating the migration mechanisms (e.g., paths, energy barriers, and related factors) would substantially bolster the credibility of the manuscript.
2. Despite conducting various CN@NiCoS heterostructures, the construction of these structures looks somewhat arbitrary. It is crucial to exercise caution and precision in constructing the slab models, especially concerning the selection of crystal facets. Numerous studies underscore the direct impact of crystal facets on HER performance. Therefore, considering the well-established impact of crystal facets

on HER performance, a systematic and meticulous experimental approach is essential to precisely identify the catalytic surface.

3. The manuscript overlooks the effect of the solvated environment and pH effect. This limitation should be checked and discussed in the manuscript. Discussing how the solvent environment might impact the heterostructure's behavior under realistic operational conditions will enhance the comprehensiveness and novelty of this manuscript.

4. Delving into the charge distribution within the heterojunction and the heterojunction with the CN overlayer would significantly enhance the understanding of the exceptional HER performance observed.

5. The manuscript asserts that the presence of a CN layer alters the characteristics of the heterojunction, consequently enhancing the HER catalytic performance. It is well-established that CN layers exhibit strong catalytic properties, characterized by a lower water dissociation energy barrier and optimal hydrogen adsorption free energy. Furthermore, the authors state that the heterojunction is encapsulated by a CN overlayer. Thus, it prompts an intriguing question: Could the actual catalytic process involve the heterojunction modifying the electronic performance of the CN layers, rather than solely the CN overlayer optimizing the performance of the heterojunction? Elaborating on this nuanced aspect could significantly contribute to comprehending the intricate interplay between the heterojunction and the CN layer in augmenting catalytic activity.

Reviewer #3 (Remarks to the Author):

This work on hydrogen production from seawater presents an intriguing approach to address the deactivation of the hydrogen evolution reaction (HER) electrode in transition-metal sulfides (TMSs). The construction of a CN@NiCoS heterostructure electrocatalyst with a nitrogen-doped carbon (CN) shell encapsulation aims to overcome the activity-stability trade-offs in TMSs by engineering sulfur migration.

The incorporation of state-of-the-art ex situ/in situ characterizations and density functional theory calculations adds depth to the study. The proposed restructuring of the CN@NiCoS interface, along with the dynamic sulfur migration, provides valuable insights into the catalyst's behavior.

One notable aspect is the stimulation of sulfur migration through the formation of S vacancies at the Ni₃S₂-Co₉S₈ heterointerface in the NiCoS heterostructure. The subsequent capture of migrated S atoms by the CN shell prevents the dissolution of sulfides into the alkaline electrolyte. This process, facilitated by a strong C-S bond, contributes to the catalyst's stability.

The formation of S-doped CN shell and S vacancies pairing sites (S/NC@NiCoS-Vs) is highlighted as a key feature leading to enhanced HER activity. The reported ultralow overpotential in both alkaline freshwater and seawater media, as well as the remarkable long-term stability, positions the catalyst favorably compared to existing literature.

However, several critical questions and considerations arise from this work:

1. Mechanistic Understanding: While the work describes the restructuring of the CN@NiCoS interface and dynamic sulfur migration, a more in-depth mechanistic understanding of the entire process, particularly the role of S vacancies and their interaction with the CN shell, would enhance the clarity of the proposed strategy. Can you provide a detailed mechanistic understanding of how the Ni₃S₂-Co₉S₈ heterostructure stimulates sulfur vacancies formation, and how these vacancies facilitate fast electron transfer to accelerate water adsorption/dissociation?

2. Comparison with Literature: The claim that the catalyst outperforms the best-reported catalysts in the literature should be substantiated with a thorough comparison, including a discussion of the specific catalysts considered and the corresponding performance metrics. Considering the emphasis on long-term stability, how does the durability of the CN@NiCoS electrocatalyst compare to other reported catalysts over extended operating periods?

3. Scalability and Practical Application: The study emphasizes the catalyst's performance in laboratory conditions; however, considerations regarding scalability, practical implementation, and potential challenges in real-world applications are crucial for assessing the broader impact of this work.

While the formation of a strong C-S bond is mentioned as a mechanism to prevent the dissolution of sulfides into the alkaline electrolyte, could you discuss potential challenges or drawbacks associated with this process? Are there any adverse effects on the electrocatalyst's performance or stability?

Can you elaborate on how the modulation of the d-band center at the Fermi energy level, resulting from the formation of S/NC@NiCoS-Vs pairing sites, contributes to the enhanced HER activity? What specific electronic interactions are involved?

4. Reproducibility: Details regarding the reproducibility of the synthesis method and the catalyst's performance across different experimental setups or conditions would contribute to the robustness of the reported findings.

5. Environmental Impact: Given the increasing focus on sustainable technologies, it would be valuable to include an assessment of the environmental impact of the proposed catalyst, considering the materials used and potential by-products.

Looking ahead, what are the potential avenues for further improvement or optimization of the CN@NiCoS electrocatalyst, and are there specific aspects that could benefit from future research efforts?

Addressing these aspects would further strengthen the significance and applicability of the reported work in the field of hydrogen production from seawater.

Response to the reviewer's comments

Reviewer #1: Though seawater as HER feedstock has attracted considerable attention, the hydrogen production from seawater is still challenging limited by the deactivation of HER electrode owing to chlorine corrosion, salts precipitations and catalyst poisoning. This manuscript submitted by Wang et al. constructed the CN@NiCoS heterostructure electrocatalyst with nitrogen doped carbon (CN) shell encapsulation to stimulate sulfur migration and facilitate electron transfer for enhanced HER activity. Moreover, they proved that the migrated S atoms were subsequently captured by the CN shell via strong C-S bond, so that to prevent the dissolution of sulfides into alkaline electrolyte and improve the stability of catalyst in alkaline freshwater and seawater media. This work provides a new guidance to control the sulfur leaching in TMSs to maintain high activity and durability towards high-performance HER. Therefore, I recommend publishing this manuscript after minor revisions. Below is a list of specific comments and questions the authors should address before publication.

Response: We greatly appreciate for the reviewer's positive feedback and the valuable suggestions for our work. We have fully considered all the comments and responded in a point-by-point manner as follows.

Comment 1: According to the HAADF-STEM image of CN@NiCoS after HER (Fig. 4e), there are obvious atomic arrangement in Co₉S₈. So, does Ni₃S₂ undergo a similar atomic rearrangement in this process? The authors have always emphasized that the heterogeneous interface of NiCoS promotes the migration of S, so how about the atomic arrangement at the interface before and after the HER?

Response: To elucidate the atomic arrangement at Ni₃S₂ phase and NiCoS interface after HER, we have conducted additional HAADF-STEM analyses for the spent CN@NiCoS catalyst after HER. As shown in Figure R1, we can observe that both Co₉S₈ and Ni₃S₂ domains close to the interface are strongly deformed with distorted atomic rearrangement on the spent sample. Especially at the interfacial region, the dim

features of defected sites/missing atoms are clearly resolved as highlighted with dashed boxes, suggesting highly defective Co₉S₈/Ni₃S₂ interface due to S migration.

Furthermore, Figure R2 shows a direct comparison of CN@NiCoS before and after HER. Compared with the pristine sample before HER (Figure R2a), the atomic rearrangement at the interface (dashed boxes) is more evident on the spent sample (Figure R2b). Similar defect features have been previously reported in literatures. [*Nat. Commun.* 14, 7849 (2023)] This could be attributed to the formation of numerous S vacancies at the Ni₃S₂/Co₉S₈ interface, leading to a more disordered lattice and more missing atoms (dim features) at the interface. This is also in line with our DFT calculations (Figure 5a) that the formation of S vacancies at the Ni₃S₂/Co₉S₈ interface is more favorable (-0.84 eV) compared to that of Ni₃S₂ (-0.20 eV) or Co₉S₈ (-0.51 eV). Based on the above results, we believe that the migration of S atoms more likely occurs on the heterojunction interface and consequently results in atomic rearrangement of Ni₃S₂/Co₉S₈ close to the interfacial region.

To address this point, we have added Figure R1 as Figure 4f in the “Revised manuscript” and included Figure R2 as Supplementary Figure 12 in “Revised Supplementary file”. We have also modified the text accordingly in the revised manuscript.

“Additionally, the Ni₃S₂ (-111) also exhibits obviously deformation with similar atomic rearrangement. Especially at the Ni₃S₂/Co₉S₈ interfacial region, the dim features of defected sites/missing atoms are clearly resolved as highlighted with dashed boxes, suggesting highly defective Ni₃S₂/Co₉S₈ interface (Fig. 4f and Supplementary Fig. 12) compared to the fresh sample before HER (Fig. 1d). This reflects the preferential formation of S vacancies at the Ni₃S₂/Co₉S₈ interface, leading to a more disordered lattice and missing atoms (dim features) at the interface⁶².” (line 24-29, page 11)

“This is consistent with our STEM observation of the defective Ni₃S₂/Co₉S₈ interface on CN@NiCoS after HER (Fig.4f and Supplementary Fig. 12).” (line 4-5, page 14)

Figure R1. HAADF-STEM images of CN@NiCoS after HER stability test. (revised Figure 4f)

Figure R2. STEM images of CN@NiCoS before HER (Figure 1d) and after HER. (revised Supplementary Figure 12).

Comment 2: In fact, the interaction of the interface between Ni_3S_2 and Co_9S_8 is not the same as the Co incorporation into Ni_3S_2 phase (line 131, page 6 and line 29, page 12), please seriously consider the difference between the interface and doping and make corresponding modifications.

Response: We greatly thank the reviewer for pointing this out. To exclude misinterpretation, we have replaced “the Co incorporation” by “the formation of $\text{Ni}_3\text{S}_2/\text{Co}_9\text{S}_8$ heterojunction” accordingly in the revised manuscript

“Notably, ...strongly infers the electron transfer from Ni sites to adjacent S likely due to the formation of $\text{Ni}_3\text{S}_2/\text{Co}_9\text{S}_8$ heterojunction.” (line 5-6, page 6)

“We can thus conclude that the formation of $\text{Ni}_3\text{S}_2/\text{Co}_9\text{S}_8$ heterojunction in CN@NiCoS stimulates the S migration...” (line 13-14, page 12).

Comment 3: It is suggested that the authors supplement the structural characterization and electrochemical data of CN@Co₉S₈ to further confirm the role of the heterogeneous interface.

Response: According to reviewer's suggestion, we have synthesized the CN@CoS catalyst for comparison and performed a full set of structural characterization and electrochemical measurements (Figure R3 and R4).

Briefly, the CN@CoS catalyst was synthesized using a similar hydrothermal method followed with sulfidation treatment, as for CN@NiCoS. The encapsulated CN@CoS structure was confirmed by XRD (Figure R3a) and TEM imaging (Figure R3b), showing a typical Co₉S₈ nanoparticle (JCPDS No.73-1442) with carbon encapsulation layers of approximately 1 nm in thickness.

HER performance of CN@CoS and CN@NiCoS electrocatalysts were evaluated in 1 M KOH solution for comparison. As indicated in Figure R4a, CN@CoS exhibits an overpotentials of 58.9, 156 and 295 mV to achieve the current density of 10, 100 and 1000 mA cm⁻², much higher than that of CN@NiCoS (4.6, 83.9 and 236 mV, respectively). This strongly infers the superiority of Ni₃S₂/Co₉S₈ interface by substantially reducing the overpotential for the HER process.

Additionally, the CN@NiCoS also exhibits superior catalytic performance of smaller Tafel slope, smaller interfacial charge transfer resistance (*R_{ct}*), and larger electrochemical double-layer capacitance (*C_{dl}*), compared with CN@CoS (Figure R4b-d). Based on all above-mentioned results, we can thus attribute the excellent electrocatalytic HER performance of CN@NiCoS to the Ni₃S₂/Co₉S₈ heterojunction (interfacial mixing/coupling), by promoting more active sites, better conductivity, and thereby reducing the energy barrier for water dissociation.

To address this point, we have included the full data set of CN@CoS in the "Revised manuscript" and "Revised supplementary file", and modified the text accordingly. Figure R3 and Figure R4 were also added as Supplementary Figure 3 and Supplementary Figure 4 in "Revised supplementary file".

"CN@CoS (single Co₉S₈ phase)" (line 1, page 5), "CN@CoS (58.9 mV)" (line 17, page 6), and "CN@CoS (53.8 mV dec⁻¹)" (line 21, page 6).

Figure R3. Structural characterization of CN@CoS electrocatalyst. (a) XRD profile. (b) HRTEM image. (revised Supplementary Figure 3)

Figure R4. Electrochemical performance of CN@CoS and CN@NiCoS electrocatalyst in 1 M KOH solution. (a) LSV curves with iR-corrected. (b) Tafel plots. (c) EIS plots. (d) Calculated electrochemical double-layer capacitance by CV curves with different rates from 20 to 100 mV s^{-1} . (revised Supplementary Figure 4)

Comment 4: There are some vague description and typo errors in this article. Thus, for a better readability, please check throughout the whole article text and correct all the errors.

Response: We greatly appreciate the reviewer's suggestion. We have thoroughly revised the article and corrected the typos for better readability. All changes in the text are marked in red.

Response to the reviewer's comments

Reviewer #2: The manuscript introduces a new strategy employing a CN@NiCoS heterostructure electrocatalyst with a nitrogen-doped carbon shell encapsulation, demonstrating remarkable HER performance. The topic is interesting, however, the exploration of the theoretical mechanism appears immature.

Response: We would like to thank the reviewer for the positive feedback and constructive comments on our work. Per reviewer's suggestion, we have performed additional DFT calculations to better elucidate the theoretical mechanism. See our detailed responses as follows.

Comment 1: The dynamics of sulfur (S) migration seemingly exert a pivotal influence on HER performance. It is imperative to provide a more profound insight into S migration, encompassing the migration pathways and energy barriers associated with S migration within the CN@NiCoS heterostructure. A comprehensive analysis elucidating the migration mechanisms (e.g., paths, energy barriers, and related factors) would substantially bolster the credibility of the manuscript.

Response: According to reviewer's suggestion, we have conducted additional DFT calculations for the energy barrier of S migration on Ni₃S₂, Co₉S₈ and NiCoS, respectively. As displayed in Figure R5, the Ni₃S₂/Co₉S₈ interface exhibit the smallest energy barrier, manifesting more favorable sulfur migration on the NiCoS heterojunction.

Specifically, the activation and breaking of metal-S bond (S2) initially occurs, followed by the adsorption of free S atoms at the metal site (S3). The adsorbed S atoms then directly occupy adjacent defect sites, leaving an S vacancy at the original position (S4). They subsequently migrate across the NiCoS (S5 and S6) and is finally trapped by the CN shell by forming S-C bond (S7). Note that the lattice mismatch/rearrangement at the Ni₃S₂/Co₉S₈ interface favors S vacancies formation (Figure 4d-f, and Figure 5a) and thus facilitates the S migration along the heterojunction.

To fully address this point, we have included Figure R5 as Figure 5 b-c in “Revised manuscript”, along with a paragraph of corresponding discussion for S migration pathways in the “Revised manuscript”.

“To provide a more profound insight into the S migration mechanism, we investigated the possibility of multiple migration pathways and their corresponding energy barriers for Ni₃S₂, Co₉S₈ and NiCoS (Fig. 5b-c). The migration of S atoms at the Ni₃S₂/Co₉S₈ interface exhibit the smallest migration energy barrier along the path, manifesting more favorable sulfur migration on the NiCoS heterojunction. As depicted in Fig. 5b, the activation and breaking of metal-S bond (S2) initially occurs, followed by the adsorption of free S atoms at the metal site (S3). The adsorbed S atoms then directly occupy adjacent defect sites, leaving an S vacancy at the original position (S4). They subsequently migrate across the NiCoS (S5 and S6) and is finally trapped by the CN shell by forming a S-C bond (S7). During this process, the lattice mismatch/rearrangement at the Ni₃S₂/Co₉S₈ interface favors S vacancies formation (Fig. 4d-f and Fig. 5a) and thus facilitates the S migration along the heterojunction.” (line 6-16, page 14)

The detailed calculations method was also added in the method section as “Ion migration energy was located by the nudged elastic band (NEB) method⁷⁵. In the NEB method, the path between the initial and final state was discretized into a series of intermediate states. The corresponding optimized geometries were relaxed until the perpendicular forces were smaller than 0.05 eV/Å.” (line 19-23, page 18).

Figure R5. (a) Reaction pathways and energy barrier of sulfur migration from the position of different sulfides. (b) corresponding Sulfur migration trajectory in CN@NiCoS. (revised Figure 5b and 5c)

Comment 2: Despite conducting various CN@NiCoS heterostructures, the construction of these structures looks somewhat arbitrary. It is crucial to exercise caution and precision in constructing the slab models, especially concerning the selection of crystal facets. Numerous studies underscore the direct impact of crystal facets on HER performance. Therefore, considering the well-established impact of crystal facets on HER performance, a systematic and meticulous experimental approach is essential to precisely identify the catalytic surface.

Response: We greatly thank the reviewer for this comment.

As the reviewer pointed out, crystal facets are crucial for the hydrogen evolution reaction (HER). Here in this paper, we constructed our DFT model using Ni₃S₂ (-110) and Co₉S₈ (331) facets based on our experimental observations. Transmission Electron Microscopy (TEM) and X-ray Diffraction (XRD) analyses (Figure R6) showed that the characteristic crystal facets are Ni₃S₂ (-110) and Co₉S₈ (331) that are mostly observed on CN@NiCoS experimentally.

Strictly following these experimental findings, we constructed the Ni₃S₂(-110)/Co₉S₈(331) interface model and achieved a stable catalyst model through full-atom relaxation optimization. The interface structure and energy levels were adjusted to minimize the energy difference between the two phases. Further DFT calculations for adsorption and catalytic reactions at the interface validated that the interface formed between Ni₃S₂ (-110) and Co₉S₈ (331) of unique electronic and structural characteristics is responsible for the high-performance HER.

To fully address this point, we have added a detailed description on the construction of DFT model and the choice of facet in the “Revised manuscript”.

“The structural model for NiCoS heterostructure was constructed using Ni₃S₂ (-110) and Co₉S₈ (331) facets according to experimental observations (Supplementary Fig. S1)”. (line 4-5, page 18)

Figure R6. (a) XRD pattern of CN@NiCoS/NF (Figure 1a). (b) HRTEM image of CN@NiCoS. (c) TEM image of NiCoS (Supplementary Figure 1b). (d) TEM image of CN@NiCoS (Supplementary Figure 1c).

Comment 3: The manuscript overlooks the effect of the solvated environment and pH effect. This limitation should be checked and discussed in the manuscript. Discussing how the solvent environment might impact the heterostructure's behavior under realistic operational conditions will enhance the comprehensiveness and novelty of this manuscript.

Response: We thank the reviewer for this valuable comment. Per reviewer's suggestion, we have performed the electrochemical performance testing under various solution environments and pH conditions (Figure R7 and R8), and achieve the impact of the solvation environment and pH conditions on the HER performance of CN@NiCoS catalyst.

Regarding the effect of solvated environment, we have investigated the impact of different alkali cations (MOH, $M^+ = Li^+, Na^+$ and K^+) on the HER performance of CN@NiCoS, while keeping a constant pH value around 14. Figure R7a illustrates that the catalytic activity for HER follows the order $K^+ > Na^+ > Li^+$ across the entire potential

range, being consistent with previous reports. [*J. Am. Chem. Soc.* 144, 1589-1602 (2021)] Tafel slopes (Figure R7b) and R_{ct} values (Figure R7c) decrease in the reverse order $\text{Li}^+ > \text{Na}^+ > \text{K}^+$, indicating that an increasing HER kinetic process and fast charge/ion transfer rate in KOH solution. This enhancement can thus be attributed to a higher concentration of weakly hydrated K^+ cation near electrode surface, altering the transport rate of cation during the HER process, that further could reduce the energy barrier of water adsorption/dissociation. [*Angew. Chem. Intern. Ed.* 60, e202102803 (2021)] Additionally, the CN@NiCoS catalyst also shows the best excellent long-term stability in KOH electrolyte (Figure R7d). This observation further demonstrates the fast charge/ion transport rate and efficient H_2 evolution capabilities of the CN@NiCoS catalyst in KOH solution.

To elucidate the pH effect, we further examined the electrochemical performance of CN@NiCoS with different pH values under alkaline KOH conditions (Figure R8). As shown in Figure R8a-b, the HER activity rises first with increasing pH value from 0.1 M to 1 M, and dramatically drops when further increasing the pH to 6 M. This can be attributed to the improved electrical conductivity due to the increased ion concentration at low pH, whereas the hydrogen binding energy (HBE) plays a dominant role in higher pH alkaline media that further impede the HER activity. [*Nat. Commun.* 10, 4876 (2019)] Based on the above-mentioned results, we can achieve the best HER activity of our CN@NiCoS catalyst in a 1 M KOH solution, by optimizing both solvation environment and pH conditions.

To address this point, we have put an additional discussion on the impact of solvation environment and pH conditions in the “Revised manuscript”. We also adder Figure R7 as Supplementary Figure 8 and Figure R8 as Supplementary Figure 9 in the “Revised supplementary file” along with supplementary notes underneath for the description of the data.

“To further demonstrate the potential of the CN@NiCoS catalyst in practical operating conditions, we evaluated the electrochemical performance under various solvation environments and pH conditions (Supplementary Fig. 8 and 9)⁵³⁻⁵⁵, and achieved the optimal HER activity of our CN@NiCoS catalyst in a 1 M KOH solution.”

(line 5-8, page 10)

Figure R7. Electrochemical performance of CN@NiCoS at pH 14 in different electrolytes with 1 M LiOH, 1 M NaOH and 1 M KOH. (a) LSV curves with iR-corrected. (b) Tafel plots. (c) EIS plots. (d) The chronoamperometry curves of 100 mA cm⁻² current density under 20 h stability tests. (revised Supplementary Figure 8)

Figure R8. Electrochemical performance of CN@NiCoS at KOH solution with different pH values (pH = 0.1 M, 1 M, 2 M, 4 M and 6 M). (a) LSV curves with iR-corrected. (b) Tafel plots. (c) Calculated electrochemical double-layer capacitance by CV curves with different rates from 20 to 100 mV s⁻¹. (revised Supplementary Figure 9)

Comment 4: Delving into the charge distribution within the heterojunction and the heterojunction with the CN overlayer would significantly enhance the understanding of the exceptional HER performance observed.

Response: We thank the reviewer for this constructive suggestion. We have performed

the charge density analysis of both NiCoS and CN@NiCoS heterojunction. As shown in Figure R9a, electrons are locally accumulated at the interfacial region of NiCoS. The higher electron density interface can notably decrease H* adsorption energy that facilitates HER activity. This is further supported by our DFT results that NiCoS exhibits a higher DOS near the Fermi level and lower H* adsorption energy (-0.14 eV) compared to Ni₃S₂ and Co₉S₈ (Fig. 5h and Supplementary Figure 15 in the original manuscript).

Furthermore, with CN overlayers, the charge density difference of CN@NiCoS (Figure R9b) demonstrates the accumulation of electrons in between NiCoS and CN layer. This suggests an enhanced electronic interaction that promotes the electron transfer from NiCoS to CN overlayers, and thus favors the H* adsorption at the CN@NiCoS heterojunction as the catalytically active sites for HER.

To address this issue, we have added the Figure R9 as Figure 5d and 5e. The interpretation of the results can be found in the “Revised manuscript”.

“As depicted in Fig. 5d-e, the charge density difference clearly shows the electron accumulation along the NiCoS interface and CN overlayer. This suggests an enhanced electronic interaction that promotes the electron transfer from NiCoS to CN overlayers. The electron enrichment along the NiCoS interface and CN overlayer facilitate the adsorption of HER intermediates as the catalytically active sites for HER⁴².” (line 17-21, page14).

Figure R9. Charge density difference of NiCoS (a) and CN@NiCoS (b). The isosurface value is $0.01 \text{ e}/\text{\AA}^3$, where green and red contours represent the electron accumulation and loss, respectively. (revised Figure 5d and 5e)

Comment 5: The manuscript asserts that the presence of a CN layer alters the characteristics of the heterojunction, consequently enhancing the HER catalytic performance. It is well-established that CN layers exhibit strong catalytic properties, characterized by a lower water dissociation energy barrier and optimal hydrogen adsorption free energy. Furthermore, the authors state that the heterojunction is encapsulated by a CN overlayer. Thus, it prompts an intriguing question: Could the actual catalytic process involve the heterojunction modifying the electronic performance of the CN layers, rather than solely the CN overlayer optimizing the performance of the heterojunction? Elaborating on this nuanced aspect could significantly contribute to comprehending the intricate interplay between the heterojunction and the CN layer in augmenting catalytic activity.

Response: We thank the reviewer for the constructive comment. To exclude the contribution of CN layer for HER activity, we prepared the CN catalyst alone and performed the HER performance of CN, NiCoS and CN@NiCoS catalyst, as shown in Figure R10 for comparison.

The comparison of LSV curve in Figure R10a clearly reveals that the CN alone exhibits almost no HER activity, and the enhancement of HER activity of NiCoS by CN encapsulation strongly suggests the promotional effect of CN overlayer that modulates the electronic structure of NiCoS heterojunction (See our detailed DFT calculations in the response to the former Question #4). CN overlayer thus serves as an electronic promotor that significantly reduced the overpotential and charge transfer resistance for enhanced HER activity of NiCoS heterojunction.

To address this point, we have added the Figure R10 as Supplementary Figure 6. In the “Revised manuscript”, we elaborately explained the promotional role of the CN overlayer in the CN@NiCoS catalyst.

“Notably, the CN overlayer alone exhibits almost no HER activity (Supplementary Fig. 6). The enhancement of HER activity of NiCoS by CN encapsulation strongly suggests the promotional effect of CN overlayer that modulates the electronic structure of NiCoS heterojunction for enhanced HER activity.” (line 2-5, page8).

Figure R10. Electrochemical performance of N-C, NiCoS and CN@NiCoS. (a) LSV curves with iR-corrected. (b) EIS plots. (revised Supplementary Figure 6)

Response to the reviewer's comments

Reviewer #3: This work on hydrogen production from seawater presents an intriguing approach to address the deactivation of the hydrogen evolution reaction (HER) electrode in transition-metal sulfides (TMSs). The construction of a CN@NiCoS heterostructure electrocatalyst with a nitrogen-doped carbon (CN) shell encapsulation aims to overcome the activity-stability trade-offs in TMSs by engineering sulfur migration.

The incorporation of state-of-the-art ex situ/in situ characterizations and density functional theory calculations adds depth to the study. The proposed restructuring of the CN@NiCoS interface, along with the dynamic sulfur migration, provides valuable insights into the catalyst's behavior.

One notable aspect is the stimulation of sulfur migration through the formation of S vacancies at the Ni₃S₂-Co₉S₈ heterointerface in the NiCoS heterostructure. The subsequent capture of migrated S atoms by the CN shell prevents the dissolution of sulfides into the alkaline electrolyte. This process, facilitated by a strong C-S bond, contributes to the catalyst's stability.

The formation of S-doped CN shell and S vacancies pairing sites (S/NC@NiCoS-Vs) is highlighted as a key feature leading to enhanced HER activity. The reported ultralow overpotential in both alkaline freshwater and seawater media, as well as the remarkable long-term stability, positions the catalyst favorably compared to existing literature.

Response: First of all, we appreciate the positive feedback and constructive comments from the reviewer. Please find our detailed response as follows.

However, several critical questions and considerations arise from this work:

Comment 1: Mechanistic Understanding: While the work describes the restructuring of the CN@NiCoS interface and dynamic sulfur migration, a more in-depth mechanistic understanding of the entire process, particularly the role of S vacancies and their interaction with the CN shell, would enhance the clarity of the proposed strategy.

Can you provide a detailed mechanistic understanding of how the Ni₃S₂-Co₉S₈ heterostructure stimulates sulfur vacancies formation, and how these vacancies facilitate fast electron transfer to accelerate water adsorption/dissociation?

Response: We thank the reviewer for this thoughtful suggestion.

Regarding the S vacancy formation, our DFT calculations (Figure 5a) have shown that the formation of S vacancies at the Ni₃S₂/Co₉S₈ interface is more favorable (-0.84 eV) compared to that of Ni₃S₂ (-0.20 eV) or Co₉S₈ (-0.51 eV). This is caused by the existence of lattice mismatch/disorder at the Ni₃S₂/Co₉S₈ interface, which has been verified by the STEM observations experimentally (Figure 4d-f). In addition, we further calculated the energy barriers for S migration. As displayed in Figure R5, the NiCoS heterojunction exhibits the smallest energy barrier for S migration, demonstrating that the migration of S atoms is kinetically more favorable along the Ni₃S₂/Co₉S₈ interface. The lattice disorder at the Ni₃S₂/Co₉S₈ interface favors the formation of S vacancies and migration of S atoms with significantly lower energy barrier.

Regarding the role of S vacancies in HER activity, we have conducted additional DFT calculations and electrochemical tests for NiCoS and NiCoS-Vs (with/without sulfur vacancies) for comparison. The NiCoS-Vs was prepared by pre-desulfurized of NiCoS with hydrogen reduction.

The DFT calculated density of states (DOS) in Figure R11a presents a prominent shift of the *d*-band center from -1.01 eV (NiCoS) to -0.93 eV (NiCoS-Vs) upon the formation of Vs. According to the classical *d*-band theory, the higher density of *d*-electrons states of NiCoS-Vs near the Fermi level can thus result in the stronger adsorption for H* intermediate, thereby facilitating H₂O adsorption/desorption, and the higher electrical conductivity for accelerated electron transfer.

This is further validated by our catalytic testing results in Figure R11 b-f. The NiCoS-Vs catalyst exhibits lower energy barrier of H₂O adsorption/desorption and H₂ evolution (Figure R11b-c), along with smaller overpotential, lower Tafel slope and *R*_{ct} values (Figure R11d-f) compared to the NiCoS catalyst. All above-mentioned results demonstrate that the formation of S vacancies can promote more *d* electrons near the Fermi level and thus accelerates the electron transfer and the kinetics of HER.

To clarify these points, we have included Figure R5 as Figure 5b and 5c, along with a corresponding description for the S migration mechanism in the “Revised manuscript”.

“To provide a more profound insight into the S migration mechanism, we investigated the possibility of multiple migration pathways and their corresponding energy barriers for Ni₃S₂, Co₉S₈ and NiCoS (Fig. 5b-c). The migration of S atoms at the Ni₃S₂/Co₉S₈ interface exhibit the smallest migration energy barrier along the path, manifesting more favorable sulfur migration on the NiCoS heterojunction. As depicted in Fig. 5b, the activation and breaking of metal-S bond (S2) initially occurs, followed by the adsorption of free S atoms at the metal site (S3). The adsorbed S atoms then directly occupy adjacent defect sites, leaving an S vacancy at the original position (S4). They subsequently migrate across the NiCoS (S5 and S6) and is finally trapped by the CN shell by forming a S-C bond (S7). During this process, the lattice mismatch/rearrangement at the Ni₃S₂/Co₉S₈ interface favors S vacancies formation (Fig. 4d-f and Fig. 5a) and thus facilitates the S migration along the heterojunction.” (line 6-16, page 14)

“Ion migration energy was located by the nudged elastic band (NEB) method⁷⁵. In the NEB method, the path between the initial and final state was discretized into a series of intermediate states. The corresponding optimized geometries were relaxed until the perpendicular forces were smaller than 0.05 eV/Å.” (line 19-23, page 18).

We have also added Figure R11 as Supplementary Figure 16 in “Revised supplementary file” and included a corresponding text in “Revised manuscript” to clarify the role of S vacancies in HER activity.

“The role of Vs for HER activity was further elucidated by DFT calculations for NiCoS and NiCoS-Vs. By incorporating S vacancies, the energy barrier of NiCoS-Vs remarkably decreased compared with that of NiCoS (Fig. 5g-h). The DOS analysis reveals the optimal d-band center closer to the Fermi level for NiCoS-Vs (-0.93 eV) than NiCoS (-1.01 eV) that facilitates electron transfer for enhanced H adsorption⁶⁷ and thus results in fast reaction kinetics for HER as indicated by the low Tafel slope (Supplementary Fig. 16).”* (line 26-29, page 14 and line 1-2, page 15)

“...the formation of Vs in NiCoS heterostructures can regulate the orbital distributions (d band center) with higher electron states near Fermi level and introduce more coordinatively unsaturated sites” (line 2-5, page 15).

Figure R5. (a) Reaction pathways and energy barrier of sulfur migration from the position of different sulfides. (b) Corresponding Sulfur migration trajectory in CN@NiCoS. (revised Figure 5b and 5c)

Figure R 11. Density Functional Theory (DFT) calculations and electrochemical performance of NiCoS and NiCoS-Vs. (a) The density of states (DOS). (b) Water dissociation energy barrier. (c) Hydrogen evolution energy barrier. (c) LSV curves with iR-corrected. (e) Tafel plots. (f) Nyquist plots. (revised Supplementary Figure 16)

Comment 2: Comparison with Literature: The claim that the catalyst outperforms the best-reported catalysts in the literature should be substantiated with a thorough comparison, including a discussion of the specific catalysts considered and the

corresponding performance metrics. Considering the emphasis on long-term stability, how does the durability of the CN@NiCoS electrocatalyst compare to other reported catalysts over extended operating periods?

Response: Based on reviewer's suggestion, we have made a more specific statement focusing on the comparison of HER performance among transition metal (TM) based catalyst under certain conditions (a current density of 10 mA cm^{-2} in 1 M KOH) (Figure R12). Additionally, we also compare the durability of CN@NiCoS with other reported catalysts in literatures, such as MoC-Mo₂C (1000 h ; 30 mA cm^{-2}), C-Co-MoS₂ (240 h ; 100 mA cm^{-2}) and GDY/MoO₃ (120 h ; 100 mA cm^{-2}). It can be seen that our catalyst also shows outstanding durability compared to other catalysts even at a higher current density (100 mA cm^{-2}). (Figure R12b)

To clarify this point, we have updated the literatures for comparison in the revised Figure 2e-f, and revised the text with a more specific statement.

“The CN@NiCoS catalyst exhibits an ultra-low overpotential ($\eta_{10}=4.6 \text{ mV}$) with an excellent stability over 1000 h in 1 M KOH solution and a current density of 100 mA cm^{-2} , outperforming the previously reported TM-based HER catalysts in the literature, by comparing activity (e.g. A-CoB/Mxene ($\eta_{10}=15 \text{ mV}$)⁴⁸, Ni/Yb₂O₃ ($\eta_{10}=20 \text{ mV}$)⁵⁰, Ni₃Sn₂-NiSn₂O_x ($\eta_{10}=14 \text{ mV}$)²⁸, Pt_x/Co catalyst ($\eta_{10}=6.9 \text{ mV}$)⁵¹), and stability (e.g. GDY/MoO₃ (120 h ; 100 mA cm^{-2})³⁷, MoC-Mo₂C (1000 h ; 30 mA cm^{-2})³⁸, C-Co-MoS₂ (240 h ; 100 mA cm^{-2})⁴⁵” (line 13-19, page 8).

Figure R 12. The comparison of overpotentials of TM-based electrocatalysts at current density of 10 mA cm^{-2} in 1 M KOH solution (a), and stability at different current density (b) with reported values of previous HER catalysts in literatures. (revised Figure 2e-f)

Comment 3: Scalability and Practical Application: The study emphasizes the catalyst's performance in laboratory conditions; however, considerations regarding scalability, practical implementation, and potential challenges in real-world applications are crucial for assessing the broader impact of this work.

While the formation of a strong C-S bond is mentioned as a mechanism to prevent the dissolution of sulfides into the alkaline electrolyte, could you discuss potential challenges or drawbacks associated with this process? Are there any adverse effects on the electrocatalyst's performance or stability?

Response: The leaching of sulfur has been a long-standing issue for the practical implementation of transition metal-based sulfides (TMSs), that can lead to severe deactivation and potential pollution to seawater. As we proposed, CN encapsulation on TMSs can substantially prevent S leaching by forming C-S bonds and enhance the HER activity and stability simultaneously. This strategy can effectively solve the S leaching issue of TMSs for practical implementation, while we also notice some other aspects that can be further improved for future study.

(1) **Potential Environmental impact of trace of sulfur in electrolyte:** Although we can significantly reduce the loss of sulfur by 18 times by the encapsulation strategy, there is still a trace of sulfur (0.013mg/L) that dissolves into the electrolyte (Table R1) The long-term environmental impact needs to be considered. To solve this issue, modification of the CN overlayer by adjusting layer thickness or N content, is thus highly demanding for future study that can inhibit the sulfur leaching, as a possible route to reduce the S concentration to a harmless level.

(2) **Simplification of the catalyst Manufacturing process:** Currently, we are using a two-step hydrothermal-sulfidation method with increased complexity and high cost for the preparation of the CN@NiCoS catalyst. To simplify the manufacturing process, a one-step method is thus highly desirable for the future study. Meanwhile, the optimization of the catalyst structure or component is also of interest to enhance the activity and stability of TMSs catalyst for seawater electrocatalysis.

To address this point, we have added the corresponding perspective in "Revised manuscript".

“The inhibition of sulfur leaching will also benefit the practical implementation of environmentally friendly TMSs catalysts for large-scale seawater electrocatalysis.”
(line 8-10, page 16).

Table R1. Sulfur concentration in electrolyte after 100h HER (revised Supplementary Table1)

	S concentration in electrolyte (mg/L)
NiCoS	2.406
C@NiCoS	0.379
CN@NiCoS	0.133

Can you elaborate on how the modulation of the d-band center at the Fermi energy level, resulting from the formation of S/NC@NiCoS-Vs pairing sites, contributes to the enhanced HER activity? What specific electronic interactions are involved?

Response: To elucidate the contribution of S/NC@NiCoS-Vs pairing site to the enhanced HER activity, we performed additional DFT calculation for the total and partial density of NiCoS, NiCoS-Vs and S/NC@NiCoS-Vs to reveal their *d*-band structures near the Fermi level.

As displayed in Figure R13, the S/NC@NiCoS-Vs shows higher *d*-electrons density (49.5 eV) and closer *d*-band center (-0.74 eV) near Fermi level compared to NiCoS and NiCoS-Vs, which facilitates the transfer of *d*-electrons to promote H* adsorption for water splitting and thus enhances HER activity. This is further corroborated by the DFT calculations, in which S/NC@NiCoS-Vs shows the optimal free energy of H₂O dissociation/H₂ evolution (Figure 5g-h in the original manuscript). The enrichment of *d* electrons near the Fermi level is mainly attributed to the enhanced *d-d* electronic interactions, resulting in a strong overlapping/hybridization of Ni 3*d* and Co 3*d* orbitals, as indicated by Figure R13. Therefore, the formation of S/NC@NiCoS-Vs pairing sites promote *d*-electron transfer and H* adsorption by enhanced *d-d* orbital coupling.

To address this point, we have also added Figure R13 as Supplementary Figure 18 in “Revised supplementary file” and a corresponding text in “Revised manuscript”.

“The S/NC@NiCoS-Vs pairing sites enable a lower d-band center and a higher DOS near the Fermi level (Fig. 5f and Supplementary Fig. 18), promoting d-electron transfer and H* adsorption by enhanced d-d orbital coupling. S vacancies can create localized states and induce a strong d-d orbital overlap (Ni 3d and Co 3d), and S-doped C can activate more d electron states near Fermi level, thereby promoting charge transfer and H* adsorption for enhanced HER activity.” (line 17-22, page 15).

Figure R13. DFT calculation in hybridization of Ni 3d and Co 3d orbital. The density of states (DOS) of NiCoS, NiCoS-Vs and S/NC@NiCoS-Vs.

Comment 4: Reproducibility: Details regarding the reproducibility of the synthesis method and the catalyst's performance across different experimental setups or conditions would contribute to the robustness of the reported findings.

Response: To ensure the data reproducibility, we have conducted multiple experiments at same conditions. As shown in Figure R14, the CN@NiCoS catalyst exhibits excellent reproducibility in terms of activity and stability (over 1000 h), in either 1M KOH solution or 1M KOH + seawater solution.

To address this point, we have been included Figure R14 as Supplementary Figure 7 in the “Revised supplementary file”, and emphasized the reproducibility of our

experimental data in the “Revised manuscript”.

“Supplementary Fig. 7 further demonstrates the excellent reproducibility of the CN@NiCoS catalyst, in either 1M KOH solution or 1M KOH + seawater solution.”

(line 3-5, page 10)

Figure R 14. Reproducibility data of CN@NiCoS catalyst. (a) LSV curves in 1 M KOH solution. (b) EIS plots. (c) LSV curves in 1 M KOH +seawater solution. (d) Durability test in 1 M KOH solution. (revised Supplementary Figure 7)

Comment 5: Environmental Impact: Given the increasing focus on sustainable technologies, it would be valuable to include an assessment of the environmental impact of the proposed catalyst, considering the materials used and potential by-products. Looking ahead, what are the potential avenues for further improvement or optimization of the CN@NiCoS electrocatalyst, and are there specific aspects that could benefit from future research efforts? Addressing these aspects would further strengthen the significance and applicability of the reported work in the field of hydrogen production from seawater.

Response: Table R1 shows that the sulfur loss in the CN@NiCoS catalyst is 18 times lower than in NiCoS, indicating that the encapsulation of CN overlayer effectively inhibits sulfur atoms loss. However, there is still a trace of sulfur (0.013mg/L) that dissolves into the electrolyte (Table R1). The long-term environmental impact by

dissolution of trace of sulfur elements into seawater needs to be considered, particularly after scaling up the electrode for practical implementation. Modification of the CN overlayer by adjusting layer thickness or N content, is thus highly desirable in future study to further inhibit the sulfur leaching, as a possible route to reduce the S concentration to a unarmful level (or zero sulfur loss). These design improvements not only enhance the catalyst's performance but also address potential environmental hazards associated with sulfur leaching, rendering the catalyst more efficient and environmentally friendly.

To address this point, we have added the corresponding perspective in “Revised manuscript”.

“The inhibition of sulfur leaching will also benefit the practical implementation of environmentally friendly TMSs catalysts for large-scale seawater electrocatalysis.”
(line 9-11, page 16).

Table R1. Sulfur concentration in electrolyte after 100h HER (revised Supplementary Table1)

	S concentration in electrolyte (mg/L)
NiCoS	2.406
C@NiCoS	0.379
CN@NiCoS	0.133

REVIEWER COMMENTS

Reviewer #1 (Remarks to the Author):

The authors have answered all the questions, and this manuscript can now be accepted in the present form.

Reviewer #2 (Remarks to the Author):

With examining all the authors' responses to comments on the original manuscript, "Ultra-low overpotential in CN@NiCoS electrocatalyst by engineering interfacial sulfur migration for durable hydrogen evolution in seawater", I think the authors made many efforts to improve their work. While the authors have made commendable progress, still some issues in the manuscript need to be addressed prior to publication.

1. In the theoretical calculation, the ultra-soft pseudopotentials have been applied through the whole calculations. However, the accuracy of this pseudopotential seems to be insufficient. At least, it should be discussed in the manuscript.
2. It would be better to provide the details on the DFT calculations. For instance, Which atoms are allowed to relax when structural optimization is performed?
3. In term of the NEB calculation, the pathway for the S migration is quite confused. More importantly, what is the transition state. Did the author check the phantom frequency? This information should be provided in the manuscript.

Reviewer #3 (Remarks to the Author):

I am pleased to recommend the acceptance of the current version of the manuscript. The author has diligently addressed all comments and suggestions provided by the reviewers, offering detailed explanations and insights that significantly enhance the strength of the work. Their thorough attention to addressing each comment demonstrates a commitment to improving the quality and clarity of the manuscript. I am confident that the revisions made by the author have substantially strengthened the manuscript, and I am therefore pleased to recommend its acceptance for publication.

Response to the reviewer's comments

Reviewer #1: The authors have answered all the questions, and this manuscript can now be accepted in the present form.

Response: We greatly appreciate the referee's positive feedback on our manuscript. No further action is required.

Reviewer #3: I am pleased to recommend the acceptance of the current version of the manuscript. The author has diligently addressed all comments and suggestions provided by the reviewers, offering detailed explanations and insights that significantly enhance the strength of the work. Their thorough attention to addressing each comment demonstrates a commitment to improving the quality and clarity of the manuscript. I am confident that the revisions made by the author have substantially strengthened the manuscript, and I am therefore pleased to recommend its acceptance for publication.

Response: We greatly appreciate the referee's positive feedback on our manuscript. No further action is required.

Reviewer #2: With examining all the authors' responses to comments on the original manuscript, "Ultra-low overpotential in CN@NiCoS electrocatalyst by engineering interfacial sulfur migration for durable hydrogen evolution in seawater", I think the authors made many efforts to improve their work. While the authors have made commendable progress, still some issues in the manuscript need to be addressed prior to publication.

Response: We greatly appreciate for the reviewer's positive feedback and the valuable suggestions for our work. We have fully considered all the comments and responded in a point-by-point manner as follows.

Comment 1: In the theoretical calculation, the ultra-soft pseudopotentials have been applied through the whole calculations. However, the accuracy of this pseudopotential seems to be insufficient. At least, it should be discussed in the manuscript.

Response: We thank the reviewer for this comment.

We were aware that the ultrasoft pseudopotential may have limitations in accuracy, but it offers significant advantages in reducing computational costs and improving efficiency. Considering the large structure of the CN@NiCoS model containing 160 atoms and involving complex triple-phase interfaces, we have opted to use the ultrasoft pseudopotential for the relevant calculations to ensure improved computational efficiency while maintaining adequate precision. The computed results of the density of states and the Gibbs free energy for the hydrogen evolution process align perfectly with our experimental findings, displayed in Figure R1 and R2. Therefore, we conclude that the computational accuracy of the ultra-soft pseudopotentials is suitable for our relevant calculations. Other types of pseudopotentials might be considered for future work to further enhance the accuracy.

To address this point, we have added the corresponding discussion on the accuracy and the choice of ultrasoft pseudopotential in the DFT calculations section of "Revised manuscript".

"Due to the CN@NiCoS model's large size (approximately 160 atoms) and its complex triple-phase interfaces, we adopted the ultrasoft pseudopotential for our calculations, ensuring improved computational efficiency while maintaining adequate precision." (line 10-12, page 18)

Figure R1. Density Functional Theory (DFT) calculations and electrochemical performance

of NiCoS and NiCoS-Vs. (a) The density of states (DOS). (b) Water dissociation energy barrier. (c) Hydrogen evolution energy barrier. (c) LSV curves with iR-corrected. (e) Tafel plots. (f) Nyquist plots. (Supplementary Figure 16)

Figure R2. Electrocatalytic HER performance and DFT calculation of N-C and S/N-C. (a) Linear sweep polarization curves. (b) Tafel slopes. (c) Nyquist plots. (d) Density Functional Theory model of N-C and S/N-C. Reaction free-energy diagram of HER on N-C and S/N-C. (e) Water dissociation. (f) Hydrogen evolution. (Supplementary Figure 17)

Comment 2: It would be better to provide the details on the DFT calculations. For instance, Which atoms are allowed to relax when structural optimization is performed?

Response: We greatly appreciate the reviewer’s suggestion.

We have provided a detailed description of the DFT calculation section in the “Revised manuscript”. Specifically, the bottom two atomic layers were held fixed while the remaining layers were allowed to relax. We conducted relaxations on all structures to refine the atomic coordinates and cell vectors until the total forces per atom were reduced to less than 10^{-4} eV/Å, while ensuring that the deviation from the target pressure remained below 0.1 GPa.

To clarify this point, we have revised the text with a more specific statement and added optimized computational models and atomic configuration of hydrogen-containing species on Ni₃S₂, Co₉S₈, NiCoS and CN@NiCoS (Figure R3 and R4) as supplementary Figure S19-20 in the revised supplementary information.

“During structural optimizations, a $1 \times 1 \times 1$ k-point grid was utilized for sampling the Brillouin zone, allowing all atomic layers to relax. Moreover, for property of the materials calculations, the bottom two atomic layers were held fixed while the remaining layers were allowed to relax. We conducted relaxations on all structures to refine the atomic coordinates and cell vectors until the total forces per atom were reduced to less than 10^{-4} eV/Å, while ensuring that the deviation from the target pressure remained below 0.1 GPa.” (line 18-24, page 18)

Figure R3. Optimized structural models. (a) Ni_3S_2 . (b) Co_9S_8 . (c) $\text{Ni}_3\text{S}_2/\text{Co}_9\text{S}_8$. (d) CN@NiCoS . Yellow, S; Gray, Ni; Blue, Co; Brown, C. (Supplementary Figure 19)

Figure R4. Optimized atomic configuration of hydrogen-containing species (H_2O^* , $^*\text{OH-H}$, $^*\text{OH}+^*\text{H}$ and H^*). (Supplementary Figure 20)

Comment 3: In term of the NEB calculation, the pathway for the S migration is quite confused. More importantly, what is the transition state. Did the author check the phantom frequency? This information should be provided in the manuscript.

Response: We thank the reviewer for this valuable comment.

To illustrate the differences in the S migration, we added diagrams depicting three different paths of S migration on Ni_3S_2 , Co_9S_8 and NiCoS , respectively, along with the structures of their initial and final states (Figure R5). We have also provided the migration path and enlarged theoretical models from initial to final state of S migration on NiCoS (Figure R6 and R7).

The nudged elastic band (NEB) approach involves optimizing the intermediate configurations to identify the energy minimum along the reaction pathway. Each configuration finds the lowest energy possible while maintaining equal spacing to neighboring images. This constrained optimization is achieved by adding spring forces along the band between images and projecting out the force component due to the potential perpendicular to the band.

Herein, we employed the NEB method to construct the geometric structure of the migration process of sulfur atoms from the known initial state (S1) to the final state (S7), linear inserting intermediate points (S2, S3, S4, S5, S6) to obtain the intermediate stages of the reaction. After optimizing the structures of these intermediate states, we obtained the S atom migration path with the lowest energy barrier (Figure R6). To ascertain which position is the transition state in the S migration process, we conducted an examination of the imaginary frequencies at the highest energy point S4. Based on the information shown in Figure R8, the S4 positions along the three migration paths all exhibit an imaginary frequency (-5934 cm^{-1} in Ni_3S_2 ; -2731 cm^{-1} in Co_9S_8 ; -4591 cm^{-1} in $\text{Ni}_3\text{S}_2/\text{Co}_9\text{S}_8$), suggesting S4 with high-energy state and an imaginary frequency as a transition state.

To address this point, we have added Figure R3, R7 and R8 as Supplementary Figure 21, 22 and 23 in “Revised supplementary file” and put a corresponding text in “Revised manuscript” to clarify the transition state.

“...S4, which likely serves as the transition state due to the highest energy state and the presence of an imaginary frequency shown in Supplementary Fig. 23.” (line 12-14, page 14)

Figure R5. Sulfur migration trajectory in Ni_3S_2 , Co_9S_8 and NiCoS phase with the initial and final configurations. (a-c) S migration path in Ni_3S_2 phase and initial/final configurations. (d-f) S migration path in Co_9S_8 phase and initial/final configurations. (g-i) S migration path in NiCoS

phase and initial/final configurations. (Supplementary Figure 21)

Figure R6. (a) Reaction pathways and energy barrier of sulfur migration from the position of different sulfides. (b) Corresponding Sulfur migration trajectory in CN@NiCoS. (revised Figure 5b and 5c)

Figure R7. Structural model of sulfur migration from initial S1 to final states S7 in NiCoS interface. (Supplementary Figure 22)

Figure R8. The transition state structure and corresponding imaginary frequency during sulfur migration process. (Supplementary Figure 23)

REVIEWERS' COMMENTS

Reviewer #2 (Remarks to the Author):

The authors have well responded with additional results and discussions. Now I agree with the publication of this work in Nature Communications.